# Learning to Grasp Anything by Playing with Random Toys

**Dantong Niu**[1,*], **Yuvan Sharma**[1,*], **Baifeng Shi**[1,*],
**Rachel Ding**[1], **Matteo Gioia**[2,3], **Haoru Xue**[1], **Henry Tsai**[1], **Konstantinos Kallidromitis**[4], **Anirudh Pai**[1],
**Shankar Sastry**[1,†], **Trevor Darrell**[1,†], **Jitendra Malik**[1,†], **Roei Herzig**[1,†]

[1]University of California, Berkeley
[2]Sapienza University of Rome
[3]ItalAI
[4]Panasonic

## Abstract

Robotic manipulation policies often struggle to generalize to novel objects, limiting their real-world utility. In contrast, cognitive science suggests that children develop generalizable dexterous manipulation skills by mastering a small set of simple toys and then applying that knowledge to more complex items. Inspired by this, we study if similar generalization capabilities can also be achieved by robots. Our results indicate robots can learn generalizable grasping using **randomly assembled objects that are composed from just four shape primitives**—spheres, cuboids, cylinders, and rings. We show that training on these "toys" **enables robust generalization to real-world objects**, yielding strong zero-shot performance. Crucially, we find the key to this generalization is an object-centric visual representation induced by our proposed detection pooling mechanism. Evaluated in both simulation and on physical robots, our model achieves a 67% real-world grasping success rate on the YCB dataset, outperforming state-of-the-art approaches that rely on substantially more in-domain data. We further study how zero-shot generalization performance scales by varying the number and diversity of training toys and the demonstrations per toy. We believe this work offers a promising path to scalable and generalizable learning in robotic manipulation. Demonstration videos, code, checkpoints and our dataset are available on our project page: https://lego-grasp.github.io/.

## 1 Introduction

*"Treat nature by means of the cylinder, the sphere, the cone, everything brought into proper perspective."*

Paul Cézanne

Robotic manipulation policies have recently achieved impressive progress, solving complex tasks in domains such as dexterous manipulation (Kumar et al., 2016; Chen et al., 2022; Wang et al., 2024; Chen et al., 2023; Qin et al., 2021), robust sim-to-real transfer (Chukwurah et al., 2024; Pinel et al., 2023; Ho et al., 2020), and long-horizon planning for multi-step tasks (Mishra et al., 2023; Simeonov et al., 2020; Pertsch et al., 2020). Yet, a fundamental challenge remains: they often fail to generalize their manipulation skills to novel objects, limiting their practical application. In stark contrast, humans show astonishing generalization capabilities in dexterous manipulation. For example, cognitive literature (Schneiberg et al., 2002; Oztop et al., 2004; Rochat, 1989; Thelen et al., 1993; Needham et al., 2002; Ruff, 1984; Bonaiuto & Arbib, 2015) suggests that children can learn to grasp by mastering only a small set of simple toys and then applying that skill to unseen complex objects. This raises a central question: *can robotic manipulation policies generalize similarly?*

---

* Equal contribution.
† Equal advising.

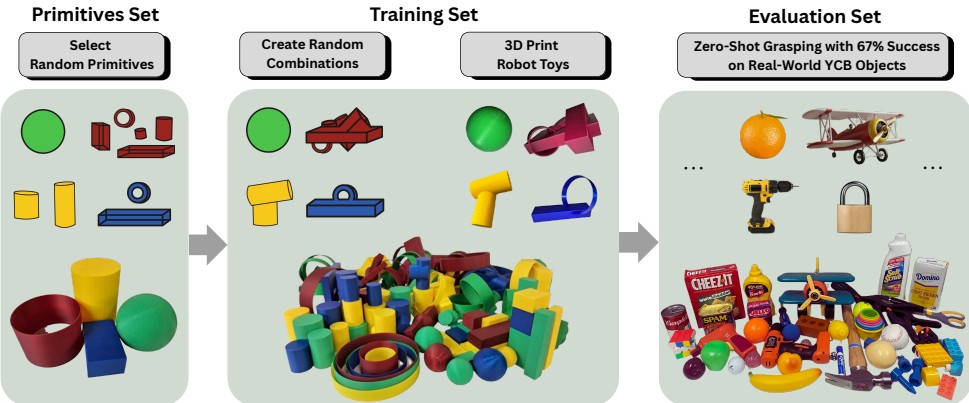

Figure 1: Our grasping policy built from just four basic primitives (left), trained exclusively on random toy compositions (middle), zero-shot generalizes to real-world objects (right) and achieves a 67% success rate on 64 objects from the YCB dataset.

In this work, we demonstrate that robots can learn to grasp novel real-world objects when trained only on *randomly constructed toys*. The design of these toys is inspired by a classic insight from Cézanne: that complex objects can be deconstructed into a vocabulary of simple shape primitives. Specifically, we construct our toys as random compositions of just four shape primitives: spheres, cuboids, cylinders, and rings. These "Cézanne toys" preserve the compositional structure of real objects while remaining sufficiently out-of-distribution, providing a challenging yet principled testbed for generalization. Trained on these random toys, our policy learns to grasp complex, unseen real-world objects in a zero-shot manner. See Figure 1 for an overview.

The key to this generalization capability, as we empirically show, lies in the usage of *object-centric visual representations*. Specifically, we introduce detection pooling (DetPool) to obtain an object-centric visual representation. This method first uses a mask of the target object to constrain the vision encoder's attention to the object region, and then applies mean pooling on the output features corresponding to the object patches. In this way, we ensure the final vision representation only contains information about the object and not the background or other distractors. We find this visual representation is the key to enable a grasping policy to generalize between the vastly different objects in training and testing. We name our framework LEGO (*LEarning to Grasp from tOys*).

To evaluate our model's generalization capabilities, we conduct a comprehensive experimental evaluation. First, we test its zero-shot performance: trained on a small dataset of 250 "Cézanne toys" with 1,500 demonstrations, our policy achieves a 67% success rate on 64 real-world YCB (Calli et al., 2015) objects, significantly outperforming larger, state-of-the-art models like OpenVLA-OFT (Kim et al., 2025) and $\pi_0$-FAST (Black et al., 2024; Pertsch et al., 2025) that are pretrained on much more data. Second, detailed ablations confirm that the key to this success is the object-centric representation induced by our DetPool mechanism, which significantly outperforms standard pooling baselines. Furthermore, we conduct thorough scaling experiments, finding that the zero-shot generalization performance scales with both toy diversity and the number of demonstrations, with the latter being more critical. Finally, we show this generalization capability is robust across robot diverse embodiments, including simple grippers and dexterous hands.

## 2 A Cézanne Toy Grasping Dataset

To evaluate the generalization of robotic grasping policies, we explore a challenging zero-shot setting: training policies exclusively on a set of out-of-distribution (OOD) objects and testing on common real-world objects. To this end, we develop a systematic approach for generating a diverse set of random, OOD objects. We draw inspiration from Cézanne's classic idea that complex objects can be abstracted into compositions of simple shape primitives. We thus generate our training objects by randomly combining these primitives. This process efficiently creates a training set of "Cézanne toys" composed of random primitives, which ensures they are OOD, while still retaining structural

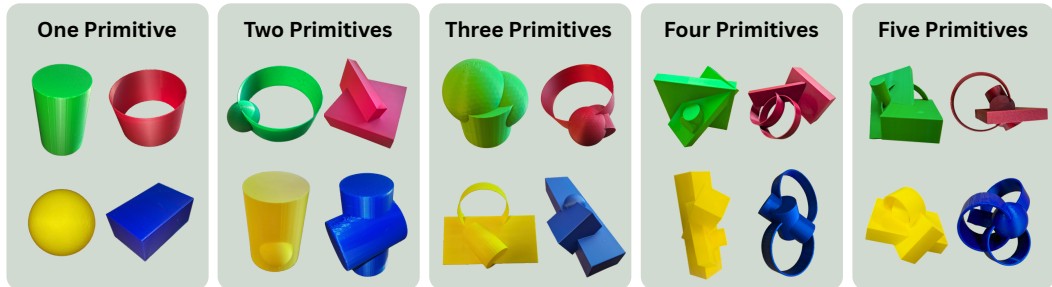

Figure 2: **Our Cézanne toys are composed of different number of primitives.** We generate each toy by randomly assembling 1-5 primitives and randomizing dimensions and colors.

and compositional properties that enable generalization. An overview is presented in Figure 1. Next, we detail our primitives' designs, the toy generation process, and the resulting grasping dataset.

**Designing the Primitives**. Inspired by prior literature (Marr & Nishihara, 1978; Tulsiani et al., 2016; Li et al., 2019), we choose four primitive types: spheres, cuboids, cylinders, and rings (See the left column of Figure 2). The primitives' scale is randomized within specific ranges. Cuboids range from 2–7.2 cm in width, 1–20 cm in height, and 2–28 cm in length; spheres range from 1–8 cm in diameter; cylinders range from 4–7 cm in diameter and 4–12 cm in height; and rings range from 6–20 cm in diameter, 0.6–1.8 cm in wall thickness, and 2–6 cm in height.

**Generating Cézanne Toys**. We generate Cézanne toys by randomly combining the primitives. Figure 2 illustrates some examples of the generated toys. We start by choosing a random number of primitives, ranging from 1 to 5. We then randomly select the required number of primitives from the four basic types, allowing repetitions, and the dimensions of each instance are randomized. The sampled primitives are then sequentially assembled to form the final toy. Specifically, the first primitive is placed at the origin, and the centroid of each subsequent primitive is randomly positioned within a previous primitive. This ensures the primitives overlap and form a coherent structure rather than scattered components. Each primitive is also assigned a random 3D rotation. Finally, the toy is assigned one of four colors: blue, red, green or yellow. By repeating this process, we generate a training set of 250 diverse toys, including 27 made of two primitives, 35 of three, 38 of four, and 47 of five, as well as individual primitives such as 46 cuboids, 18 balls, 20 cylinders, and 19 rings. All toys are both simulated and 3D printed for grasping data collection.

**Collecting Grasping Data**. We collect toy grasping trajectories in both simulation and real. In simulation, we use ManiSkill (Tao et al., 2025) with a Franka arm and gripper; in the real world, we use the same Franka arm with a Robotiq gripper, as well as a Unitree H1-2 humanoid equipped with Inspire RH56DFTP hands. We collect all data via teleoperation, except for grasping individual primitives in simulation, which is performed using motion planning. During collection, we ensure a diverse set of grasping poses per object, since individual objects can be grasped in many different ways. We collect 2,500 trajectories in simulation, 1,500 on the real Franka, and 500 on the H1-2.

## 3 THE LEGO METHOD

To enable a policy trained on our "Cézanne toys" to generalize to real-world objects, we introduce a novel object-centric approach. Our method's key distinction from full-scene architectures is its use of a detection pooling mechanism to obtain an object-centric visual representation, which we empirically show is the key to robust generalization. This section details our preliminaries (Section 3.1), full architecture (Section 3.2), and the detection pooling method (Section 3.3).

### 3.1 PRELIMINARIES

**Robotic Tasks**. Robotic tasks can be represented as temporal sequences of observations and actions. The observations typically consist of visual observations $i_{1:T}$ and proprioceptive states $s_{1:T}$, where $T$ is the episode length, $i_t \in \mathbb{R}^{N \times H \times W \times 3}$ denotes the images captured by $N$ cameras at time step

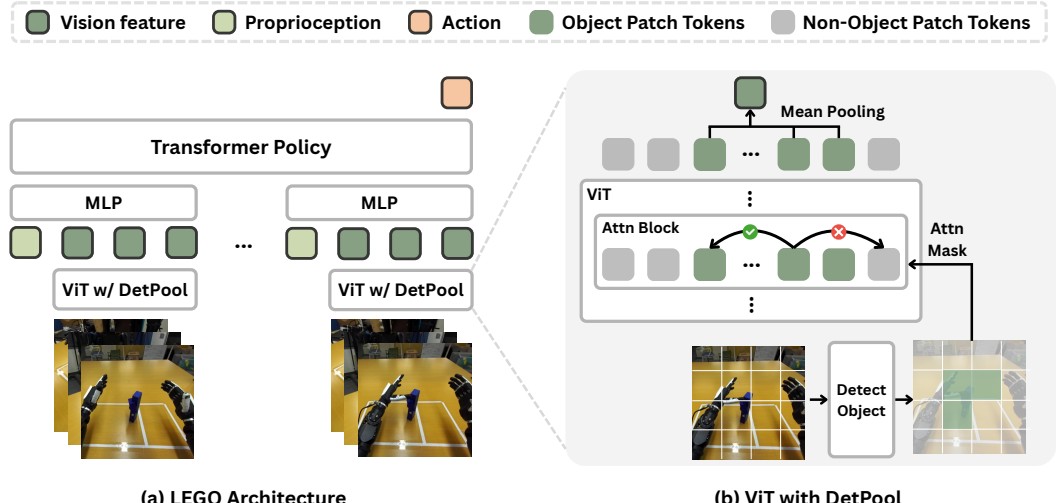

Figure 3: **The LEGO architecture with DetPool.** **(a)** LEGO uses a ViT with DetPool to extract features of the target object and uses a transformer to predict future actions based on the visual features and the proprioception. **(b)** The ViT extracts features that focus on the target object via DetPool which restrains the attention to the object patches using an attention mask and performs mean pooling on the output object patch tokens to get the final object-centric vision feature.

$t$, and $s_t \in \mathbb{R}^{d_s}$ is the proprioception (e.g., the joint positions) of the robot at step $t$. The actions $a_{1:T} \in \mathbb{R}^{d_a}$ represent how the robot commands its joints (e.g., target joint positions) at step $t$.

**Policy Learning**. The training objective is to learn a policy that maps a history of the past $C$ steps— visual inputs $i_{t-C+1:t}$ and robot states $s_{t-C+1:t}$—to a future action sequence of length $K$, in order to successfully complete the task: $\pi(i_{t-C+1:t}, s_{t-C+1:t}) \rightarrow a_{t:t+K-1}$.

## 3.2 ARCHITECTURE

Below, we describe the different components of our policy architecture as well as the training objective. Figure 3 (a) illustrates the overall LEGO architecture.

**Vision Encoder**. Given the observations $i_{t-C+1:t}$ and $s_{t-C+1:t}$ from past $C$ steps, our model uses a vision encoder to encode each set of visual observations $i_t$ into visual embeddings $e_t^{1:N}$, where $e_t^n \in \mathbb{R}^{d_e}$ is the embedding of the $n$-th camera image at step $t$, and $d_e$ is the hidden dimension of the vision encoder. We use a pretrained MVP (Xiao et al., 2022) as the vision encoder. These resulting features are then input into a transformer-based architecture for further processing.

**Transformer Policy**. We use a transformer-based architecture as our main policy network. It first concatenates the visual embeddings $e_t^{1:N}$ and the proprioception $s_t$ along the channel dimension into a single token, and then projects it with an MLP. The transformer backbone then takes the projected tokens from all past $C$ steps and predicts the concatenated actions $a_{t:t+K-1}$ for the next $K$ steps from the last token. The transformer is designed to have the same size as a ViT-B (Dosovitskiy et al., 2021) to get the best performance (see the ablation in Section 4.5).

**Training Objective**. Following the regular behavior cloning algorithm, the training loss is the mean $\ell_1$ loss between the predicted actions $\hat{a}_{t:t+K-1}$ and the ground-truth actions $a_{t:t+K-1}$, i.e., $\mathcal{L} = \frac{1}{Kd_a}\|\hat{a}_{t:t+K-1} - a_{t:t+K-1}\|_1$.

To enable generalization to novel objects, we design the vision encoder to be object-centric through detection pooling, introduced in the next section.

## 3.3 DETECTION POOLING

We design a detection pooling mechanism in the vision encoder such that the extracted visual feature is focused on the object to be grasped, as shown in Figure 3 (b). Specifically, we first obtain the

object segmentation mask for each frame using SAM 2 (Ravi et al., 2024b). We then use the object mask to set the attention mask in the vision encoder such that there is no attention between object patch tokens and non-object patch tokens. In this way, we ensure that the object patch tokens only contain features from the object itself while ignoring features from non-object patch tokens. Note that this method still allows the vision encoder to understand where the object is in the scene due to the use of positional embeddings. At the end of the vision encoder, we obtain the object-centric visual feature by applying mean pooling on the object patch tokens, which is the final visual embedding we use for the policy model. We empirically find that DetPool is crucial for achieving strong zero-shot generalization compared to other pooling methods such as mean and attention pooling that do not restrict the attention mask within the ViT and only pool the final output tokens (Section 4.2).

## 4 EXPERIMENTS

We evaluate LEGO on the YCB object benchmark (Calli et al., 2015) using the ManiSkill simulator (Tao et al., 2025). For comparison, we include vision-language-action (VLA) models such as $\pi_0$-FAST and OpenVLA-OFT, which aim to generalize through large-scale pretraining. We further analyze how performance scales with the number of unique toys and demonstrations. Beyond simulation, we test LEGO on two real-world setups: a 7-DoF Franka Emika Panda with a 1-DoF Robotiq 2F-85 adaptive gripper, where evaluation is done on the YCB benchmark, and an Unitree H1-2 humanoid with Inspire dexterous hands, evaluated on a 13-object set of everyday items.

### 4.1 IMPLEMENTATION DETAILS

**Model and Training Setup**. LEGO is implemented using PyTorch (Paszke et al., 2019). Its architecture consists of a ViT-L encoder from MVP (Xiao et al., 2022) for feature extraction and a ViT-Base transformer backbone. The policy is conditioned on a history of $C = 16$ timesteps to predict $K = 16$ future actions. For our DetPool mechanism, we use SAM 2 (Ravi et al., 2024a) to obtain object masks for real-world images and use ground-truth masks in simulation. The model is trained on eight NVIDIA A6000 GPUs using the AdamW optimizer (Loshchilov & Hutter, 2019), and evaluated on a single A6000.

**State and Action Parameterization**. We parameterize the proprioceptive space using the joint angles of the robot arm used and a continuous gripper state (when applicable). This yields an 8-dimensional vector for the Franka setup, and a 40-dimensional vector for our H1-2 setup (which includes feedforward torques and finger joints). The model then conditions on state vectors from past timesteps and predicts action vectors for future timesteps, where state and action vectors are represented using absolute joint angles, rather than relative (delta) angles.

### 4.2 SIMULATION EVALUATION

**Experimental Setup**. Our training set contains 2,500 demonstrations, comprising 10 successful grasps for each of our 250 unique toys. To analyze scaling laws, all models are also trained on subsets of this data. For evaluation, we use a set of 65 graspable objects from the YCB benchmark, selecting only those feasibly graspable by the Franka robot; each object is tested 16 times on a predefined grid, and we report the mean success rate across all trials.

**Baselines**. We compare LEGO (86M parameters) against two significantly larger, state-of-the-art VLAs that rely on large-scale pretraining: $\pi_0$-FAST (3B) (Black et al., 2024) and OpenVLA-OFT (7B) (Kim et al., 2025). $\pi_0$-FAST is a state-of-the-art VLA model trained on a large-scale robotics dataset. OpenVLA-OFT is a 7B-parameter VLA model pretrained on the Open-X Embodiment (OXE) dataset (Collaboration et al., 2023). Both models are fine-tuned on the same data as ours. To validate the contribution of our core DetPool mechanism, we also conduct ablation studies, replacing DetPool with standard alternatives like attention pooling, CLS pooling, and mean pooling.

**Results**. Our simulation results, summarized in Table 1, highlight the superior generalization and scalability of LEGO compared to baselines. While LEGO's performance scales reliably with more data—achieving a top success rate of 80% with 2,500 demonstrations, the state-of-the-art VLA baselines falter. We find that $\pi_0$-FAST is too data-hungry for the small dataset and struggles with a real-to-sim domain gap from its pretraining. Similarly, OpenVLA-OFT shows initial promise with

Table 1: **Results of zero-shot grasping in simulation.** We compare our model with state-of-the-art models (OpenVLA-OFT and $\pi_0$-FAST) finetuned on our dataset in simulation, as well as different pooling baselines. Our model outperforms the finetuned baselines in simulation, with our DetPool proving key to generalization by boosting performance 22-48% over other pooling baselines.

| Method | # Demos | | | | | |
|---|---|---|---|---|---|---|
| | 250 | 500 | 1000 | 1500 | 2000 | 2500 |
| OpenVLA-OFT (Kim et al., 2025) | 30.10 | 36.35 | 22.31 | 15.38 | 14.71 | 12.79 |
| $\pi_0$-FAST (Black et al., 2024) | 8.85 | 7.60 | 7.69 | 8.56 | 4.23 | 4.13 |
| Ours - Attn Pooling | 34.71 | 40.10 | 44.23 | 48.27 | 49.81 | 51.63 |
| Ours - CLS Pooling | 24.71 | 20.29 | 36.92 | 41.44 | 42.40 | 49.81 |
| Ours - Mean Pooling | 32.98 | 30.38 | 36.15 | 39.90 | 40.29 | 40.58 |
| Ours - Det Pooling | **56.63** | **68.17** | **71.15** | **74.62** | **76.83** | **80.00** |

250–500 demonstrations but quickly overfits as more data is added, causing its performance to deteriorate. On the other hand, while attention pooling is the strongest baseline, it is still significantly outperformed by our DetPool mechanism. In contrast, DetPool enables robust and scalable generalization, underscoring the effectiveness of object-centric visual representation for generalizability.

### 4.3 FRANKA ROBOT EVALUATION

**Experimental Setup**. For real-world experiments, we use a 7-DoF Franka Emika Panda arm with a Robotiq 2F-85 gripper, consistent with the DROID benchmark. We 3D-print the 250 toys with the highest simulated success rates in simulation and collect 1,500 successful grasp demonstrations. All models are then evaluated on a test set of 64 YCB objects. Following the simulation protocol, each object is tested 16 times on a predefined grid, and we report the mean success rate.

**Baselines**. We compare LEGO with strong baselines including $\pi_0$-FAST, OpenVLA-OFT, and ShapeGrasp (Li et al., 2024b) which is a training-free, LLM-based approach that uses pretrained language models to decompose objects geometrically before selecting a graspable part.

A common theme across these methods is reliance on large-scale pretraining: either extensive in-domain robot trajectories (for $\pi_0$-FAST) or internet-scale multimodal data (for ShapeGrasp). In contrast, LEGO is trained from scratch on only 1,500 demonstrations, yet achieves competitive performance despite being orders of magnitude smaller in both dataset size and model scale.

**Results**. As shown in Table 2, LEGO achieves the second-best performance among all models tested, highlighting the effectiveness of our approach. It outperforms OpenVLA-OFT and ShapeGrasp, as well as $\pi_0$-FAST in its zero-shot setting, despite these being much larger models and $\pi_0$-FAST using in-domain DROID data. This also shows the strength of our object-centric representation, as LEGO attains superior performance using far less data and a smaller model architecture.

The finetuned version of $\pi_0$-FAST achieves the best overall performance, which we hypothesize is because finetuning on additional demonstrations from our DROID setup allows it to utilize its pretrained knowledge and adapt to the specific lighting and physical environment, improving performance. In addition, finetuning on the randomized toys dataset likely provides an additional performance boost. In contrast, OpenVLA-OFT is less effective. We observed that minor inaccuracies frequently caused grasp failures, indicating difficulty in generalizing to novel objects and settings.

### 4.4 H1-2 DEXTEROUS HANDS EVALUATION

**Experimental Setup**. We perform real-world experiments with the Unitree H1-2, a humanoid robot. Each 7-DoF arm is equipped with a 6-DoF Inspire RH56DFTP hand, which has 12 total joints. The 6 DoFs capture the independent motions of the fingers, while the 12 joints result from each DoF being implemented as a pair of mechanically linked joints driven by a single linear servo. This hand design mimics human-like dexterity more closely than traditional gripper end-effectors, making it well-suited for experiments that require fine-grained grasping.

Table 2: **Zero-shot grasping results on the real Franka robot.** We compare our model against ShapeGrasp, OpenVLA-OFT (finetuned), and state-of-the-art $\pi_0$-FAST (zero-shot and finetuned). Our model achieves a 66.67% success rate, outperforming all baselines except finetuned $\pi_0$-FAST.

| Method | Pretraining | Tuned on Toys | # Parameters | # Demos |
| --- | --- | --- | --- | --- |
| | | | | 1500 |
| OpenVLA-OFT (Kim et al., 2025) | OXE | ✔ | 7B | 9.47 |
| $\pi_0$-FAST (Black et al., 2024) | $\pi$ Dataset + 75K DROID | ✘ | 3B | 61.82 |
| $\pi_0$-FAST (Black et al., 2024) | $\pi$ Dataset + 75K DROID | ✔ | 3B | **76.56** |
| ShapeGrasp (Li et al., 2024b) | GPT4o | ✘ | - | 26.56 |
| Ours | | ✘ | 86M | 66.67 |

Table 3: **Results on H1-2 humanoid robot.** We compare our model with state-of-the-art models ($\pi_0$-Fast and OpenVLA-OFT) that are finetuned on our data. We show our model achieves superior performance without any pretraining on real objects.

| Method | Bell Pepper | Pink Cube | Baton Cookies | Solder Coil | Tomato | Pink Ribbed Ball | Piggles Stuffed Toy |
| --- | --- | --- | --- | --- | --- | --- | --- |
| OpenVLA-OFT | 0 | 0 | 40 | 20 | 20 | 0 | 40 |
| $\pi_0$-FAST | 20 | 20 | 0 | 20 | 20 | 20 | 40 |
| Ours | **60** | **40** | **60** | **40** | **60** | **60** | **60** |
| | Mike Wazowski Stuffed Toy | Red Tape Dispenser | Red Solo Cup | Paper Towel Roll | Hand Sanitizer | Yo-Yo | Average |
| OpenVLA-OFT | 60 | 0 | 0 | 60 | 0 | 0 | 18.46 |
| $\pi_0$-FAST | 60 | 40 | 40 | 40 | 20 | 0 | 26.15 |
| Ours | **60** | **60** | **20** | **60** | **60** | **20** | **50.77** |

We evaluate our model and baselines on 13 everyday objects using the left arm and hand. Each object is tested 5 times across a predefined grid, with trials scored identically to the Franka experiments.

**Baselines**. We compare LEGO with OpenVLA-OFT and $\pi_0$-FAST (see Section 4.3 for details).

**Results**. As shown in Table 3, LEGO achieves the highest success rate of 50.77% in a more challenging setting than the Franka DROID experiments, despite being trained from scratch with only 500 demonstrations. In contrast, $\pi_0$-FAST struggles due to limited demonstrations and the likely absence of this embodiment in its pretraining data. OpenVLA-OFT also underperforms, having been pretrained on robot arm and gripper data without humanoid or dexterous-hand examples. These results underscore LEGO 's data efficiency and the role of DetPool in enabling robust generalization.

### 4.5 ABLATION STUDIES

We present our ablation studies below, conducted within the ManiSkill simulation environment, which offers a stable and reproducible setting for rigorously evaluating the impact of various factors.

**Effect of Number and Diversity of Demonstrations**. We perform an ablation study to examine how both the number of unique toys in the training set and the number of grasping demonstrations influence performance. Specifically, we construct eight object sets containing 1, 5, 15, 25, 125, 250, 500, and 1000 unique toys, respectively. For each set, we collect 2,500 total grasping demonstrations and train our model using varying numbers of demonstrations per set. The results (left panel, Figure 4) indicate that increasing the number of unique objects improves performance, but with diminishing returns. In contrast, the number of demonstrations has a stronger impact on learning generalizable grasping, a result consistent with findings from cognitive science literature (Smith & Slone, 2017).

**Effect of Model Size**. To investigate how the size of the policy's transformer backbone affects performance, we conduct an ablation study. Using the 250-object set—which yields the best overall performance—we vary the transformer's size and evaluate the policy across different numbers of

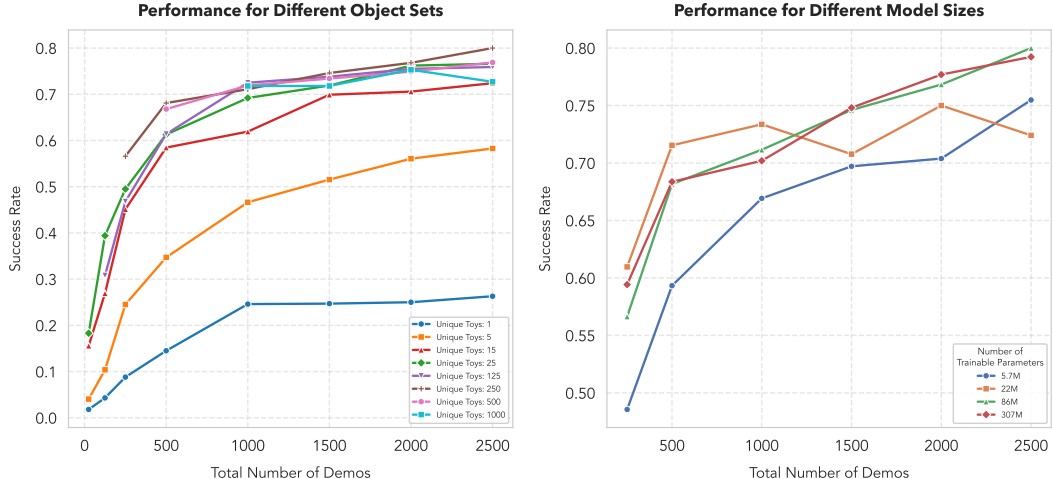

Figure 4: **Scaling studies. Left**: The zero-shot success rate scales with both the number of demos and the number of unique toys. We also find that once the number of demos is sufficient, 15 toys is already enough to achieve a robust zero-shot transfer. **Right**: The performances scales with the size of the policy transformer until it saturates at the size of 86M.

demonstrations. The results, shown in the right panel of Figure 4, indicate that ViT-Base (86M) is the best overall choice: it matches or slightly surpasses ViT-Large (307M) in performance while being significantly smaller and thus allowing for faster inference.

**Importance of Individual Primitives**. To assess the relative importance of each of the four primitives, we conduct an ablation study in which the training set excludes toys containing a given primitive. For each case, the model is trained with varying numbers of demonstrations. The results, presented in Table 4, show that the sphere is the most critical primitive, as its exclusion results in the largest performance degradation. In contrast, the ring and cylinder appear to be less important, with a relatively small performance drop when they are omitted.

**Effect of Toy Complexity**. We perform an ablation study to measure the relationship between toy complexity, quantified by the number of constituent primitives, and model performance. The model is trained on demonstrations containing toys with 2-5 primitives. As shown in Table 5, toys with two primitives contribute the most to performance, while toys with five primitives are still beneficial but less influential. This is likely due to the evaluation set's distribution of sizes, which contains more toys with two or three distinct parts; highly complex toys with five parts are relatively rare.

Table 4: **Ablation of primitive types.** We study the importance of each primitive type by removing each one from the primitive set, and report the mean success rate on the YCB object set for different numbers of total demonstrations.

| Primitive Removed | 100 | 200 | 500 | 1000 |
|---|---|---|---|---|
| Cuboid | **37.88** | 56.35 | 65.38 | 72.12 |
| Sphere | 44.13 | **47.31** | **61.83** | **63.08** |
| Ring | 44.23 | **67.5** | 68.56 | **72.6** |
| Cylinder | **45.29** | 57.6 | **69.52** | 72.31 |

Table 5: **Ablation of toy complexity.** We study the importance of each toy complexity level by training polices only on toys composed of a certain number of primitives, and report the mean success rate on the YCB object set for different numbers of total demonstrations.

| Toy Complexity | 25 | 125 | 250 |
|---|---|---|---|
| Two Primitives | **9.04** | **32.6** | **44.42** |
| Three Primitives | 7.31 | 15.77 | 23.17 |
| Four Primitives | 7.69 | 12.4 | 23.36 |
| Five Primitives | 4.32 | 10.87 | 10.19 |

**Geometric Similarity Between Training Toys and Target Objects**. We analyze the geometric similarity between the training toys and target objects by computing the Chamfer distance (CD)

Table 6: **Geometric similarity analysis between training toys and target objects.** Objects are grouped into difficulty buckets based on the 33rd and 66th percentiles of Chamfer distance (CD) to the closest training object. We report average success rate (mean $\pm$ 95% confidence interval) and summary statistics of Chamfer distance within each bucket.

| Difficulty | # Objects | Success | $\text{CD}_{Mean}$ | $\text{CD}_{Std}$ | $\text{CD}_{Min}$ | $\text{CD}_{Max}$ |
|---|---|---|---|---|---|---|
| Easy | 22 | $75.6 \pm 13.8$ | 0.0568 | 0.0123 | 0.0392 | 0.0756 |
| Medium | 21 | $83.0 \pm 12.0$ | 0.0990 | 0.0070 | 0.0871 | 0.1080 |
| Hard | 22 | $81.5 \pm 12.7$ | 0.1300 | 0.0283 | 0.1082 | 0.2164 |

between each target object and its closest counterpart in the training set. Target objects are grouped into *easy*, *medium*, and *hard* categories based on the 33rd and 66th percentiles of Chamfer distance.

As expected, the average Chamfer distance increases monotonically across difficulty buckets. However, as shown in Table 6, the average success rate remains relatively consistent across groups. This indicates that grasp performance does not strongly depend on geometric similarity to the training toys. Overall, the results suggest that our approach is robust to geometric variation and exhibits strong object-level generalization.

## 5 RELATED WORK

**Cognitive Approaches for Manipulation**. Developmental psychology shows that infants acquire manipulation skills through exploration and practice (Thelen et al., 1993; Schneiberg et al., 2002; Needham et al., 2002). Several studies (Ruff, 1984; Rochat, 1989; Yoshida & Smith, 2008) have revealed a gradual progression in which they learn to manipulate objects by focusing on increasingly diverse features such as shape and color. Rakison & Butterworth (1998) further demonstrate that infants generalize to unseen objects by applying learned actions to familiar object parts. Motivated by this literature, we investigate achieving similar generalization in robotic manipulation.

Existing object modeling, inspired by infant learning (Farhadi et al., 2009), uses descriptive attributes (Cohen et al., 2019; Sun et al., 2013), explicit segmentation (Liu et al., 2024; Li et al., 2024a;b; Vahrenkamp et al., 2016; Aleotti & Caselli, 2011), or represents objects as 3D primitives (Tulsiani et al., 2016; Monnier et al., 2023; Lin et al., 2025). Our work builds on these ideas and explores whether generalized object representations can emerge from just a few primitives.

**Generalization in Robotic Manipulation**. While several robotic manipulation models have shown capability of mastering various real-world tasks (Zhao et al., 2023; Fu et al., 2024; Barreiros et al., 2025), they often generalize poorly to novel objects and scenes. One common approach to address this is scaling up the training data (Brohan et al., 2022; Zitkovich et al., 2023; Intelligence et al., 2025; Eppner et al., 2021; Fang et al., 2020; Ye et al., 2025). In contrast, we show that strong generalization capabilities are still achievable even with a few hours of training data. Another line of works improves the generalizability of robotic models through heavy data augmentation (Hansen & Wang, 2021; Tobin et al., 2017; Sadeghi & Levine, 2016), while we achieve strong generalization without any augmentation. Other works enhance transferability by learning better visual representations (Burns et al., 2023; Srirama et al., 2024). Compared to these approaches, which usually require costly visual pretraining, we improve generalizability by obtaining object-centric visual representations via DetPool, which is light-weight and can be applied to any pretrained vision model.

**Object-Centric Vision Models**. Object-centric models represent a scene as a collection of discrete object-level entities. They are shown to improve performance and robustness in computer vision. Methods like Burgess et al. (2019); Engelcke et al. (2019); Locatello et al. (2020) learn object-centric representations from 2D images, typically for scene decomposition. Extending object-centric learning to the temporal domain, Herzig et al. (2021) introduce an object-region transformer that learns temporally coherent object-level features across video frames for tasks such as action recognition and dynamics prediction. Other approaches extend this idea into the 3D domain via world models (Ferraro et al., 2023; Jeong et al., 2025). Unlike these works, our approach focuses on the effect of object-centric representations on generalizable robotic manipulation.

When it comes to manipulation, a variety of object-centric grasping approaches have been explored (Chen et al., 2024; Zurbrügg et al., 2024; Mandikal & Grauman, 2020; Emukpere et al., 2025), including with large VLA models (Zhong et al., 2025a) pretrained with internet-scale information. In contrast, our model is orders of magnitude smaller, focusing specifically on grasping to demonstrate that the visual representation induced by detection pooling drives object-level generalization. Papers such as Devin et al. (2017) address generalizable robot learning through object-centric methods by using attention mechanisms. However, they only evaluate the generalization between similar objects and do not explore the limit of generalization between vastly different objects such as real objects. The most similar approach to ours is OTTER (Huang et al., 2025), which uses the vision-language attention map in CLIP to obtain object-centric visual representations. However, it is only limited to CLIP, while our method can be applied to any vision transformer.

## 6 Conclusion

In this work, we demonstrate that robots can acquire robust general-purpose grasping skills by learning from a simple set of objects composed from just four basic shape primitives: spheres, cuboids, cylinders, and rings. We show that training on these toys enables a policy to generalize to a wide range of real-world objects. Our method learns an object-centric visual representation using a detection pooling and transformer architecture, and is trained on a dataset of 250 toys with 1,500 demonstrations in the real Franka setting. This policy achieves a 67% zero-shot success rate on the YCB dataset, outperforming state-of-the-art models such as $\pi_0$-FAST and OpenVLA-OFT despite them being trained on more diverse and larger datasets. Our findings on grasping scaling laws highlight how we can efficiently optimize performance with limited data. Ultimately, this work demonstrates a scalable path to robotic manipulation by showing that real-world grasping generalization can emerge from learning on object composites of a few primitive shapes. We believe this work offers a promising path to scalable and generalizable learning in robotic manipulation.

## 7 Limitations and Future Work

While we show that our method offers a promising path toward generalized grasping, it is important to acknowledge its limitations to guide future research. One key limitation is the diversity of the training domain: the model's performance may degrade on objects with different physical properties. Furthermore, our current work focuses on simple, single-step grasping. Future work could extend this approach to complex, long-horizon tasks such as cloth folding and manipulation in dynamic scenes. Finally, the computational cost of the model's architecture presents a challenge for real-world deployment on resource-constrained hardware, pointing toward a need for future optimization.

### Acknowledgments

We would like to thank Caitlin Regan for setting up teleoperation on the Unitree H1-2, Hunter Tsai for his efforts in building a 3D printing farm and supplying the robot toys for real experiments, and Raj Saravanan for helping with initial toy design and data collection with teleoperation. We would like to thank Ilija Radosavovic for helpful discussions and feedback. We also thank Nicole Walters for creating our lovely logo. Authors, as part of their affiliation with UC Berkeley, were supported in part by the Defense Advanced Research Projects Agency (DARPA) under the TIA-MAT program (HR00112490425), Design of Robustly Implementable Autonomous and Intelligent Machines, and/or the Berkeley Artificial Intelligence Research (BAIR) industrial alliance program, as well as the Humanoid Intelligence Center program. Matteo Gioia conducted this work while enrolled in the Data Science Doctorate at Sapienza University of Rome, funded by the European Union—NextGenerationEU, Mission 4, Component 1 (CUP B53C22003870006). Matteo Gioia also acknowledges partial support from Sapienza grant RM1241910E01F571 (V3LI).

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

## A  Toys Design

### A.1  Real 3D Toys Design and Manufacturing

**Primitive Design**. To design the toys, we wrote a Python script that uses the SAPIEN physics engine to generate random dimensions for a set of primitives in the amount desired, such as a cuboid and a cylinder for a two primitive toy. These primitives are assembled into a toy by placing them at random offsets between them that ensure the primitives are still physically connected to each other. Finally, we export the toy mesh into an STL file using the Trimesh library. We list out the dimension ranges of the primitives in Table 7.

Table 7: Dimension ranges for primitive shapes.

| Shape | Diameter/Width (cm) | Height (cm) | Length (cm) |
|---|---|---|---|
| Cuboid | 2–7.2 | 1–20 | 2–28 |
| Sphere | 1–8 | N/A | N/A |
| Cylinder | 4–7 | 4–12 | N/A |
| Ring | 6–20 | 2–6 | 0.6–1.8 (wall thickness) |

**Toy Manufacturing**. We printed a total of 250 toys in PLA filament, in addition to multiple test prints to validate the toy geometry and print quality. This was done using a fleet of eight Bambu P1P printers over a span of four weeks, enabling a maximum throughput of 200 toys per week by printing multiple toys on a single print bed (excluding FivePrimitive toys, whose size meant that they took up the entire print bed and took significantly longer to print). The fleet was managed using the Bambu Farm Manager platform.

The biggest challenge with printing the toys was the delicate geometry of the rings. The original designs had very thin ring walls that would snap during removal from the print bed. To compensate, we redesigned the toys to have thicker ring walls to strengthen the print. In addition, the intersection of shape primitives often resulted in large overhanging bodies, which required large amounts of tree supports to be modelled and printed. Toys with larger primitive counts had significantly higher print times due to their increased volume and complexity. Certain FivePrimitive toys had to be scaled down in size by 20% to fit in the 256mm x 256mm x 256mm print volume.

We have provided the full Bambu printer settings used for our prints for ease of reproducibility in Tables 17, 18, 19, and 20. Any omitted settings are assumed to take the default value. Organizing the toys into boxes and using a label printer to label them with their names is important for keeping track of all the toys, such as if a reprint is needed.

## B  Robustness Analysis

We systematically study the robustness of LEGO under challenging visual conditions, including lighting variation, scene clutter, and distractor objects in the ManiSkill environment.

### B.1  Lighting Variation

To evaluate robustness to illumination changes, we reduce the lighting intensity to 70% and 40% of the original environment lighting during evaluation. As shown in Table 8, LEGO maintains nearly identical performance across lighting conditions, with only minor degradation even under severe illumination reduction. This suggests that the learned representations are largely invariant to moderate lighting variations.

### B.2  Scene Clutter

To study robustness to clutter, we add 2–5 randomly selected YCB objects to the tabletop scene in addition to the target object. Results in Table 9 show a performance drop in cluttered environments; however, LEGO remains effective despite the presence of multiple irrelevant objects. This indicates that the model can selectively attend to task-relevant regions even in visually crowded scenes.

Table 8: **Performance under different lighting conditions.**

| Lighting | 250 demos | 2500 demos |
|---|---|---|
| 100% Lighting | 56.63 | 80.00 |
| 70% Lighting | 56.44 | 79.33 |
| 40% Lighting | 55.00 | 78.65 |

Table 9: **Performance under cluttered environments.**

| Condition | 250 demos | 2500 demos |
|---|---|---|
| No Clutter | 56.63 | 80.00 |
| Clutter | 46.73 | 67.21 |

## B.3 DISTRACTOR OBJECTS

We further evaluate robustness to distractors by placing either a random object or an additional instance of the target object (lookalike distractor) on the table in the ManiSkill environment. As shown in Table 10, both distractor types reduce performance, with visually similar distractors presenting a particularly challenging scenario. Nevertheless, LEGO retains strong performance, suggesting that it can distinguish target objects even under visual ambiguity.

Table 10: **Performance under distractor settings.**

| Condition | 250 demos | 2500 demos |
|---|---|---|
| No Distractor | 56.63 | 80.00 |
| Single Random Distractor | 50.19 | 71.35 |
| Single Lookalike Distractor | 49.23 | 74.13 |

Overall, LEGO demonstrates strong robustness to lighting variation, clutter, and visually similar distractors. We attribute this robustness to the detection pooling mechanism, which encourages the model to focus on target-object regions rather than irrelevant parts of the scene.

## B.4 MASK NOISE

We additionally evaluate the robustness of our method to noise in the segmentation mask by perturbing the bounding box provided to SAM during tracking. Specifically, we introduce randomized offsets to the ground-truth bounding box and quantify the noise level using the Intersection over Union (IoU) between the original and perturbed boxes. Lower IoU corresponds to higher noise.

Table 11 reports success rates under varying IoU levels for both 250 and 2500 demonstration settings. Performance degrades only marginally as mask noise increases. Even at 60% IoU, the drop in success rate is small relative to the 100% IoU setting.

This robustness stems from detection pooling operating at the patch level. Since attention masks are applied over patches, small spatial perturbations often result in the same set of object patches being selected, leading to inherent tolerance to moderate mask inaccuracies.

Table 11: **Effect of mask noise on performance.** Noise is quantified by the IoU between the ground-truth and perturbed bounding boxes.

| IoU | 250 demos | 2500 demos |
|---|---|---|
| 100% | 56.63 | 80.00 |
| 80% | 55.83 | 79.29 |
| 60% | 53.19 | 77.98 |

## C    ADDITIONAL EXPERIMENTS

### C.1    MORE COMPARISONS OF LEGO TO OBJECT-CENTRIC APPROACHES

We further compare our method to related object-centric approaches in robotic manipulation. Specifically, we add GROOT Zhu et al. (2023) and DexGraspVLA Zhong et al. (2025b) as experimental baselines in ManiSkill, with the results presented in Table 12.

Table 12: **Comparison between LEGO and object-centric approaches.** We further include additional comparisons with object-centric approaches on a subset of our data. These methods are fine-tuned on our 250-toy and 2500-toy demonstration datasets and then evaluated in a zero-shot manner on downstream tasks.

| Method | 250 demos | 2500 demos |
|---|---|---|
| GROOT Zhu et al. (2023) | 0.87 | 1.54 |
| OpenVLA-OFT Kim et al. (2025) | 30.10 | 12.79 |
| $\pi_0$-FAST Black et al. (2024) | 8.85 | 4.13 |
| DexGraspVLA Zhong et al. (2025b) | 20.77 | 48.85 |
| **Ours** | **56.63** | **80.00** |

The first new baseline, GROOT, uses object-centric point clouds to predict robot actions. This approach can be difficult to scale and deploy in the real world, and as seen by the results, also does not generalize well from toys to real-world objects. Even with the use of ground-truth depth data from the simulator, this approach is unable to adapt when encountering novel objects. The other object-centric baseline DexGraspVLA achieves decent performance, getting higher success rates than OpenVLA-OFT and $\pi_0$-FAST. However, it still does not reach the performance levels unlocked by detection pooling, and its performance does not scale with increasing amounts of data as well.

### C.2    EFFECT OF TOY COLOR

We measure the impact of toy color on performance by conducting an ablation study comparing a policy trained on a set of only red toys to our original set where toys were randomly assigned one of four colors (red, green, blue, yellow). As shown in Table 13, color diversity improves performance. This is likely because exposure to toys with varying colors during training helps the model learn more robust visual features so it can generalize better to real-world objects.

Table 13: **Effect of toy colors on zero-shot generalization.** We compare the zero-shot performance of model trained on single-color toys with multi-color ones. Training on toys with multiple colors boost performance by about 1%-4% although training on single-color toys still yields a strong generalization to real objects.

| Toy Colors | 250 | 500 | 1000 | 1500 | 2000 | 2500 |
|---|---|---|---|---|---|---|
| Red | 50.1 | 66.44 | 68.94 | 72.6 | 75.48 | 76.35 |
| **Red + Green + Blue + Yellow** | **56.63** | **68.17** | **71.15** | **74.62** | **76.82** | **80** |

### C.3    PUSH-TO-TARGET MANIPULATION

We evaluate our method on an additional ManiSkill manipulation task, *push object to target*, which requires sustained contact and precise force control to slide an object into a designated region, and differs substantially from grasping. As in the grasping setup, we train on a randomized set of 250 unique toys and evaluate zero-shot generalization on the YCB object benchmark.

The quantitative results are reported in Table 14, with OpenVLA-OFT Kim et al. (2025) included as a baseline. Detection pooling continues to deliver superior performance in this distinct manipulation setting. Although OpenVLA-OFT shows improved results compared to its grasping performance, its success rate remains substantially lower than that of our method. Moreover, performance improves

Table 14: **Performance comparison for "push object to target" task.**

| Method | 250 Demos | 2500 Demos |
|--------|-----------|------------|
| OpenVLA | 58.46 | 75.77 |
| **Ours** | **67.31** | **81.83** |

with additional demonstrations for both approaches, consistent with the scaling trend observed in the grasping task. Overall, these findings indicate that the benefits of detection pooling extend beyond grasp-centric behaviors to more general contact-rich manipulation tasks.

## C.4 ABLATIONS ON CONDITION AND PREDICTION LENGTH

We study the effect of varying the condition length $c$ and prediction length $k$ on model performance. Results are reported in Table 15.

Table 15: **Performance as a function of condition length $c$ and prediction length $k$.** The numbered columns denote the number of demonstrations used during training. All experiments described in the main paper use $c = 16$ and $k = 16$.

| $c$ | $k$ | 250 | 500 | 1000 | 1500 | 2000 | 2500 |
|-----|-----|-----|-----|------|------|------|------|
| 16 | 4 | 63.65 | 71.92 | 63.94 | 69.85 | 69.42 | 70.67 |
| 16 | 8 | 59.04 | 69.90 | 68.85 | 77.31 | 73.27 | 78.65 |
| **16** | **16** | **56.63** | **68.17** | **71.15** | **74.62** | **76.83** | **80.00** |
| 8 | 16 | 38.94 | 54.71 | 57.60 | 62.31 | 70.00 | 71.15 |
| 4 | 16 | 22.12 | 30.38 | 38.46 | 52.07 | 52.88 | 54.30 |

From the bottom three rows in Table 15, we observe that reducing the condition length $c$ leads to a substantial drop in performance across all training steps. This indicates that sufficient conditioning context is critical for stable and effective prediction. In contrast, comparing the top three rows shows that increasing prediction length $k$ has a comparatively smaller effect. While longer prediction horizons generally improve final performance, the gains are more gradual and less dramatic than those obtained from increasing the condition length.

## C.5 DELTA VS. ABSOLUTE END-EFFECTOR CONTROL

We chose absolute joint control rather than delta control in all our experiments since we empirically found it to perform better. We present a comparison of the two control modes across a varying number of demonstrations in Table 16.

Table 16: **Delta vs. absolute joint control across different data scales.** Absolute control consistently and substantially outperforms delta control at every demonstration count.

| Method | 250 | 500 | 1000 | 1500 | 2000 | 2500 |
|--------|-----|-----|------|------|------|------|
| Delta Control | 29.71 | 30.19 | 31.63 | 34.90 | 36.44 | 37.02 |
| **Absolute Control** | **56.63** | **68.17** | **71.15** | **74.62** | **76.83** | **80.00** |

As shown in Table 16, absolute control significantly outperforms delta control across all data scales. Delta control plateaus around 30–37% success rate regardless of the number of demonstrations, suggesting that it struggles to benefit from additional training data. In contrast, absolute control not only achieves much higher performance at every scale but also exhibits a clear upward trend as the number of demonstrations increases, reaching 80.00% at 2500 demos. These results confirm that absolute joint control is the more effective control mode for our setting, likely because it provides a more stable learning target that avoids the compounding errors inherent to delta predictions.

## D    REAL ROBOT HARDWARE CONFIGURATION

### D.1    FRANKA EMIKA PANDA

We deploy our policy on a Franka Panda Robot with 7 DoFs equipped with a RobotiQ gripper and a ZED 2i wrist camera. The 7 DoFs allow for precise and dexterous manipulation of the gripper to grasp various types of objects from every part. Two additional ZED 2i cameras are positioned to the left and right sides of the robot. Each camera provides an RGB stream at 720p and 30 FPS, without depth information. The hardware configuration is shown in Figure 5.

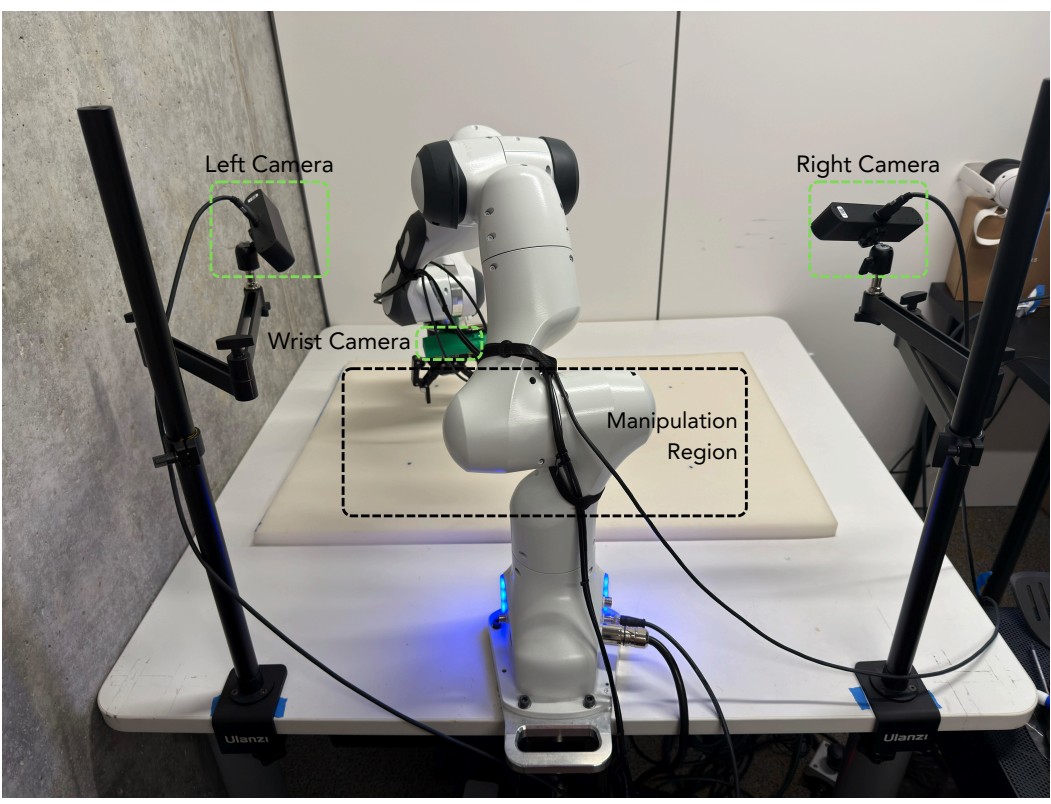

Figure 5: Hardware Configuration for Franka Emika Panda with Robotiq Gripper.

### D.2    H1-2 HUMANOID WITH DEXTEROUS HANDS

We also deploy our policy to a Unitree H1-2 humanoid robot. The robot is equipped with two Inspire RH56DFTP dexterous hands, each with 6 DoFs, 12 motors and a linear drive design with six miniature linear servo drives and six pressure sensors integrated inside. Given these characteristics, the hands are a good fit to emulate real dexterous operations by a human. The robot is also equipped with a ZED 2i head camera mounted below the original head camera to improve the quality of the egocentric data captured. Two ZED 2i cameras are positioned to the side of the robot, creating a similar setup to the one used for the Franka arm. Each camera provides an RGB stream at 720p resolution and 30 FPS, without depth information. The hardware configuration is shown in Figure 6.

## E    ROBOT DEMONSTRATIONS COLLECTION

### E.1    MANISKILL SIMULATION MOTION PLANNING

ManiSkill (Tao et al., 2025) is a simulation environment built on the SAPIEN framework. We generated a dataset of demonstrations for a Franka arm grasping and lifting single primitive objects using scripted planners.

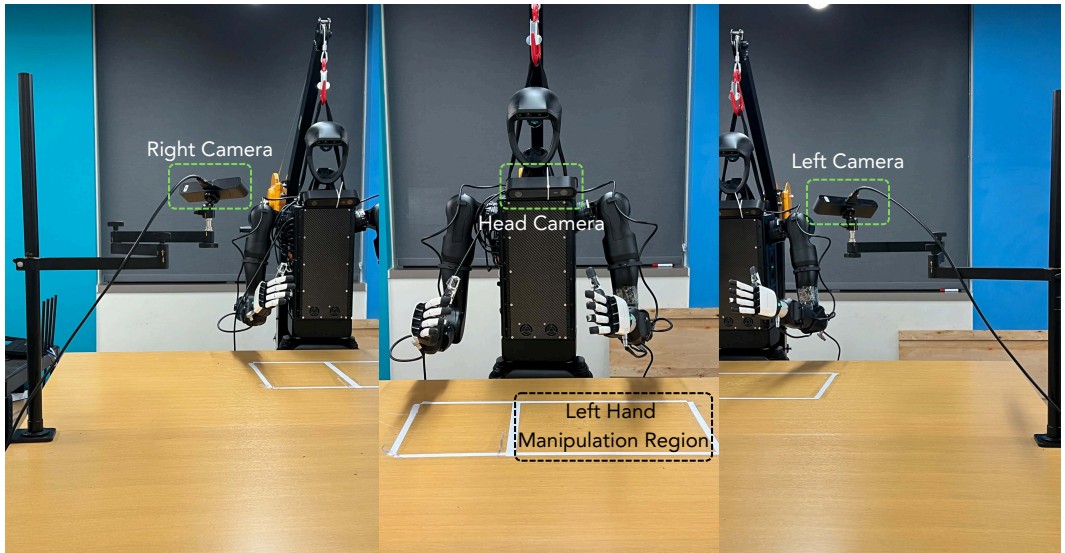

Figure 6: Hardware Configuration for H1-2 Humanoid with Inspire Dexterous Hands.

### E.2 MANISKILL SIMULATION TELEOPERATION

Using ManiSkill, we also designed a simulation environment to collect data via human teleoperation. The teleoperation data collection process was then standardized as follows: the arm was first positioned slightly above the target grasp pose, then moved down to the grasp position, its gripper was closed to secure the object, and the object was then lifted upward. While it is possible to fully automate the data generation pipeline using grasping planners for more complex toys, we encountered engineering challenges that ultimately led us to rely on teleoperation.

### E.3 FRANKA REAL ROBOT TELEOPERATION

We teleoperated the Franka robot using a Meta Quest 3 headset, with only the right-hand controller mapped to arm control. Each pick-up demonstration was executed in one smooth motion on a foam-covered table to protect the objects. We recorded videos from the left and right ZED cameras as well as the wrist camera, and additionally logged the robot's proprioceptive states. We adopt the Franka-DROID robot settings provided by the DROID dataset (Khazatsky et al., 2024).

### E.4 H1-2 WITH DEXTEROUS HANDS TELEOPERATION

To collect real-world data for the H1-2 humanoid robot, we used a teleoperation setup with the Apple Vision Pro (AVP) VR headset, built on Unitree's XR Teleoperate platform. The headset provides an RGB 2D view from the head camera, giving the operator a human-like perspective via the Vuer visualization toolkit. Our tracking script controls both dexterous hands and monitors the arms' poses; however, due to hardware limitations, we restricted data collection to the left arm and hand. Each recorded episode corresponds to a single toy-grasping demonstration.

## F DETPOOLING

**Creating Attention Masks**. To pool visual features, we first extract the target object's segmentation mask from camera views. In the ManiSkill Franka simulation, ground truth object masks are directly available and used to identify vision encoder patches overlapping with the object. For the real Franka and H1-2 dexterous hand setups, we manually annotated 200 toy images with bounding boxes to train a Faster R-CNN detector with a ResNet-101 backbone . The detector's bounding boxes are

---

https://github.com/facebookresearch/detectron2

Table 17: Print Quality Settings

| Setting | Value |
|---|---|
| Layer Height | 0.3 mm |
| Initial Layer Height | 0.3 mm |
| Line Width (All) | 0.62 mm |
| Seam Position | Aligned |
| Smart Scarf Seam Application | On |
| Scarf Application Angle | 155° |
| Scarf Steps | 10 |
| Scarf Joint for Inner Walls | On |
| Role-based Wipe Speed | On |
| Slice Gap Closing Radius | 0.049 mm |
| Resolution | 0.012 mm |
| Arc Fitting | On |
| Elephant Foot Comp. | 0.15 mm |
| Ironing Type | No Ironing |
| Initial Layer Density | 90% |

Table 18: Print Speed Settings

| Setting | Value |
|---|---|
| Initial Layer Speed | 35 mm/s |
| Initial Layer Infill | 55 mm/s |
| Outer Wall Speed | 120 mm/s |
| Inner Wall Speed | 150 mm/s |
| Top Surface Speed | 150 mm/s |
| Sparse Infill Speed | 100 mm/s |
| Travel Speed | 500 mm/s |
| Normal Printing Accel. | 10000 mm/s$^2$ |
| Travel Acceleration | 10000 mm/s$^2$ |
| Initial Layer Travel Accel. | 6000 mm/s$^2$ |
| Initial Layer Accel. | 500 mm/s$^2$ |
| Inner Wall Accel. | 0 mm/s$^2$ |
| Outer Wall Accel. | 5000 mm/s$^2$ |
| Top Surface Accel. | 2000 mm/s$^2$ |
| Sparse Infill Accel | 100% |

Table 19: Print Strength Settings

| Setting | Value |
|---|---|
| Wall Generator | Classic |
| Order of Walls | Inner/Outer |
| Bridge Flow | 1 |
| Wall Loops | 2 |
| Top/Bottom Shell Pattern | Monotonic |
| Top Shell Layers | 3 |
| Top Shell Thickness | 0.8 mm |
| Bottom Shell Layers | 3 |
| Bottom Shell Thickness | 0 mm |
| Internal Infill Pattern | Rectilinear |
| Sparse Infill Density | 10% |
| Sparse Infill Pattern | Triangles |
| Infill/Wall Overlap | 15% |
| Infill Direction | 45° |
| Ensure Vertical Shell | Enabled |

Table 20: Print Support Settings

| Setting | Value |
|---|---|
| Enable Support | On |
| Type | Tree(auto) |
| Style | Default |
| Threshold Angle | 30° |
| Remove Small Overhangs | On |
| Raft Layers | 0 |
| Top Z Distance | 0.2 mm |
| Bottom Z Distance | 0.2 mm |
| Top Interface Layers | 2 |
| Top Interface Spacing | 0.5 mm |
| Support/Object XY Distance | 0.35 mm |
| Support/Object First Layer Gap | 0.2 mm |
| Tree Support Branch Distance | 5 mm |
| Tree Support Branch Diameter | 2 mm |
| Tree Support Branch Angle | 45° |

then used as input to SAM 2 to obtain segmentation masks, from which the attention masks are constructed in the same manner as in simulation.

**Pooling Visual Features**. For detector-based pooling, we follow a standard vision processing pipeline. The image is first patchified and passed through Transformer blocks. From the final block, we obtain spatial feature maps, and then apply the attention mask obtained above to pool the corresponding spatial features, yielding the final pooled features. For the visual encoder, we adopt the off-the-shelf ViT-L MVP model, which was pre-trained with a masked autoencoder objective and has been demonstrated to be effective for robotic control in prior work (Radosavovic et al., 2023).

## G  ROBOTIC POLICY TRAINING DETAILS

**Observation**. For the simulated Franka robot setting, we use three camera views as visual inputs: two fixed cameras mounted on the tabletop and one wrist-mounted camera. For the real Franka robot, the hardware configuration follows the standard DROID setup, with two tabletop-mounted cameras and one wrist-mounted camera. For LEGO policy training, we use only the two tabletop-mounted cameras as visual inputs.

**Action Space**. The LEGO policy is conditioned on the previous and current states, represented by the 7-DoF arm joint positions and the 1-DoF gripper state. The policy is trained to predict future action chunks, consisting of joint poses and gripper states.

**Training Details**. We adopt a learning rate of $5 \times 10^{-4}$ with a weight decay of 0.01. Training is conducted for 900 epochs with a 30-epoch warm-up and a global batch size of 512. In comparison to foundation VLA models such as $\pi_0$-FAST (Black et al., 2023) and OpenVLA-OFT (Kim et al., 2024), our approach demonstrates substantially lower GPU memory requirements and achieves faster convergence, highlighting the efficiency of the proposed architecture.

# H    BASELINES IMPLEMENTATION DETAILS

## H.1    $\pi_0$-FAST

We adopt $\pi_0$-FAST (Black et al., 2024) as a baseline for our simulated Franka, real-world Franka, and real-world H1-2 Dexterous Hands experiments, following the official code and instructions  .

**Simulated Franka Robot**. On the ManiSkill simulation platform, we fully finetuned the released base autoregressive $\pi_0$-FAST model on our simulated toy dataset. We use joint position control, adapting the pretrained model to predict the absolute 7-DoF joint pose and 1-DoF gripper status. We use left camera view and wrist camera view as visual inputs, and use "pick the toy" as the language instruction. We follow the default learning rate in the original implementation and finetune the model for 10K steps with a batch size of 32 for each setting reported in Table 1.

**Real Franka Robot**. For the real-world Franka robot, we use the DROID setting. Instead of velocity control, we adopt joint position control and finetune the released base autoregressive $\pi_0$-FAST on our teleoperated toy dataset. The pretrained model is adapted to predict the absolute 7-DoF joint pose and 1-DoF gripper status. We use left camera view and wrist camera view as visual inputs, and use "pick the toy" as the language instruction. Following the default learning rate, we train for 10K steps under both the 500-demonstration and 1500-demonstration settings shown in Table 2.

**Real H1-2 Robot with Dexterous Hands**. We further extend our setting to a real humanoid platform equipped with dexterous hands. Specifically, we fine-tune the released base autoregressive $\pi_0$-FAST model on our 500-demonstration teleoperated toy dataset using delta joint control. The action space consists of a 20-dimensional vector, including 7-DoF right arm joints, 6-DoF wrist torque, and 6-DoF finger joint angles. We experiment with both absolute joint control and delta joint control. Visual observations are obtained from the left camera view and the head camera view, and the language instruction is "pick the toy with dual arms." Our results show that delta control consistently outperforms absolute control.

In this embodiment setting (in contrast to the DROID setting, which is covered during pretraining), we observe that $\pi_0$-FAST tends to overfit when fine-tuned on limited data, likely due to its large model capacity. To mitigate overfitting, we select an early checkpoint where the cross-entropy loss remains at a reasonable value above 1.0. In comparison, for the DROID setting, the model does not exhibit overfitting even when the loss decreases to around $1 \times 10^{-2}$, likely because it is pretrained on a large-scale DROID dataset. For all results reported in this paper, we use the default learning rate and train for 1K steps on the 500 demonstrations, as summarized in Table 2.

## H.2    OPENVLA-OFT

We use OpenVLA-OFT (Kim et al., 2025) as a baseline for both simulation and real-world experiments, following the official implementation and finetuning instructions . We use LoRA (Hu et al., 2021) finetuning with a rank of 32 for all experiments.

**Simulated Franka Robot**. On the ManiSkill simulation platform, we use delta joint position control and input images from the front, base, and wrist cameras. The model is trained with a global batch size of 16 and an initial learning rate of $1.25 \times 10^{-4}$, decayed to $1.25 \times 10^{-5}$ after 100,000 steps.

---

https://github.com/Physical-Intelligence/openpi
https://github.com/moojink/openvla-oft

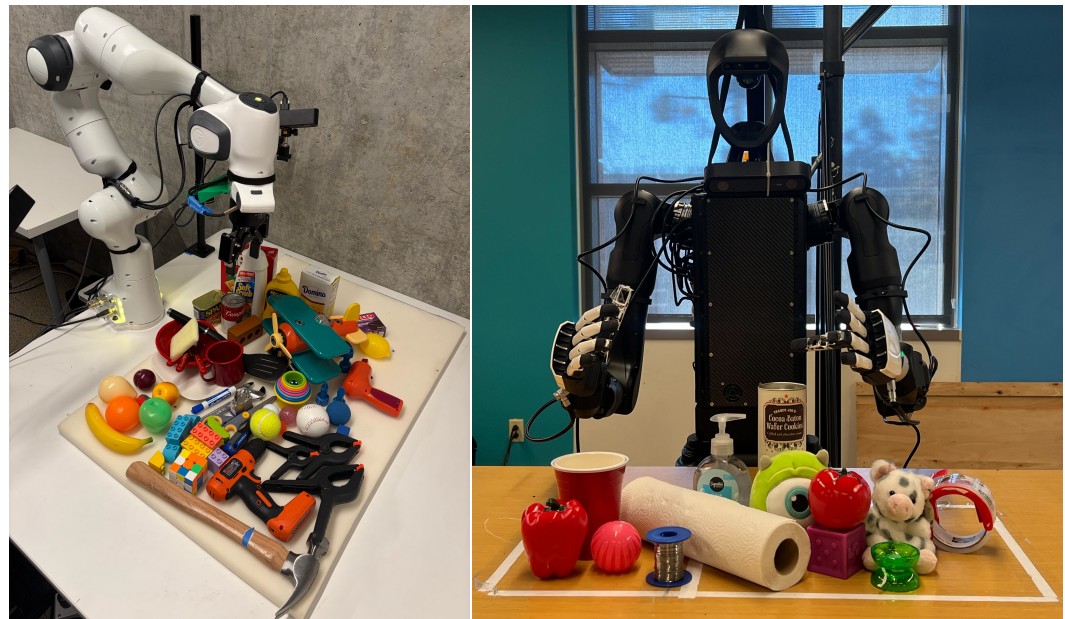

Figure 7: **Real-world Evaluation Settings.** We have DROID Franka setting with YCB dataset on the left and H1-2 robot with dexterous hands and 13 everyday objects.

Training runs for a total of 150,000 steps, with checkpoints at every 20,000 steps evaluated to select the best-performing model for each experiment.

**Real Franka Robot**. In the real Franka DROID setting, we use delta joint position control, consistent with the simulation experiments. The model receives images from the left, right, and wrist cameras. The model is trained with a global batch size of 16 and an initial learning rate of $1.25 \times 10^{-4}$, decayed to $1.25 \times 10^{-5}$ after 100,000 steps, for a total of 150,000 steps. We used the last checkpoint for evaluation.

**Real H1-2 Robot with Dexterous Hands**. The model is conditioned on images from the left, right, and head cameras. It receives a 26-dimensional state vector—corresponding to 7 DoF per arm and 6 DoF per hand—and predicts a 40-dimensional output, which includes absolute joint targets for all joints as well as feedforward torques for both arms. Training uses a global batch size of 16 and an initial learning rate of $1.25 \times 10^{-4}$, decayed to $1.25 \times 10^{-5}$ after 100,000 steps, for a total of 150,000 steps. We used the last checkpoint for evaluation.

### H.3 SHAPEGRASP

We evaluate ShapeGrasp on our real Franka setup using the official implementation . ShapeGrasp uses GPT-4o to identify a graspable part from a decomposition graph, where nodes represent object parts (modeled as convex shapes) and their spatial relationships. It outputs a pixel location along with a $z$-axis rotation for a top-down grasp. Using a calibrated Intel RealSense D435 camera, we project the pixel-level grasp prediction into 3D space. An executable grasp trajectory is then generated by interpolating between the robot's current pose and the predicted grasp pose.

## I EVALUATION DETAILS

For the simulated Franka robot, we use the default task environment "PickClutterYCB-v1" for evaluation, with details available in the official documentation. For the real-world experiments, we consider two settings, as shown in Figure 7. The left panel illustrates the standard DROID setup with the YCB dataset used for evaluation, while the right panel shows the H1-2 robot equipped with Inspired dexterous hands and the 13 everyday objects used for evaluation.

---

https://github.com/samwli/ShapeGrasp

### I.1 MANISKILL SIMULATION EVALUATION

To evaluate policies in simulation, we defined a $0.15 \times 0.15$ m square workspace, subdivided into a $4 \times 4$ grid. The grid was constructed from the Cartesian product of the sets $\{-0.075, -0.025, 0.025, 0.075\}$ along both the $x$ and $y$ axes, resulting in 16 evenly spaced placements. For each trial, the object was placed at one grid location with its $z$-rotation initialized using a random seed. Each object was tested across all 16 placements, and success rates were averaged across objects and placements. A trial was considered successful (1) if the robot lifted the object above a height threshold of 0.3 m. For OpenVLA-OFT policies, we reduced the success threshold to 0.15 m, as the gripper would often prematurely open after grasping the object for these policies. Trials in which the object was not lifted above the threshold were marked unsuccessful (0).

### I.2 FRANKA ROBOT EVALUATION

To evaluate our policy on the Franka Panda arm, we defined a $0.5 \times 0.28$ m rectangular workspace on the table, subdivided into a $4 \times 4$ grid. For each trial, the object was placed in one of the 16 grid cells, with its $z$-axis orientation randomized. We evaluated policies by testing each object across all 16 placements and averaged the results to compute the final success rate. A trial was considered successful (1) if the robot securely lifted the object above a height threshold of 0.2 m, and unsuccessful (0) otherwise.

### I.3 H1-2 HUMANOID DEXTEROUS HANDS EVALUATION

To evaluate our policy on the H1-2, we defined a grasping workspace by taping off a $40\,\text{cm} \times 36\,\text{cm}$ rectangular zone on the table, positioned within the head-mounted ZED camera's field of view and centered between the two Inspire hands. This rectangle was subdivided into six equally sized $3\,\text{in} \times 3\,\text{in}$ squares. For each object tested, we conducted five grasping trials, placing the object in a different square for each trial. Performance was scored as 1 if the robot successfully picked up the object and 0 otherwise. All trials were executed using the left arm and hand. During evaluation, we encountered technical issues with the Inspire hands—most notably unresponsive thumb joints on both sides—which limited the scope of humanoid grasping experiments we were able to carry out.

