# OpenReview forum: "Learning to Grasp Anything By Playing with Random Toys"
_ICLR.cc/2026/Conference — ICLR 2026 Poster_

### Official Review · Reviewer_V8CY · 2025-10-30

**Soundness:** 3
**Presentation:** 3
**Contribution:** 2
**Rating:** 4
**Confidence:** 3

**Summary:**

The main contribution of this paper consists of the introducing a data-efficient, object-centric learning framework—called LEGO (LEarning to Grasp from tOys)— that enables robots to generalize grasping skills to unseen real-world objects by training only on randomly assembled synthetic toys, built from only 4 primitive shapes. It gets its inspiration from an analogy between human cognitive development and robotic skill acquisition.
As many other papers in the field, it emphasizes the critical role of an object-centric perception model, through a detection pooling (DetPool) mechanism.

**Strengths:**

* Paper clearly written, well structured ans easy to understand, with good intuitive explanations
Elegant, data-efficient learning setup leveraging primitive-based object construction.
* The proposed method shows strong empirical performance with minimal data (the data efficiency clain is well demonstrated empirically).
* I appreciate the experimental coverage including some ablations and, more importantly, multiple robot platforms and real-world experiments!

**Weaknesses:**

* The approach is overall heuristic (based on an analogy with child development) and does not formally support that toy-based training will yield to generalizable features.
In robotic manipulation, it is well known that geometry is not enough: mass distribution, friction/contact forces, affordances, texture are also very important factors to deal with when we want to generalize to a wide class of objects. All these aspects are not sufficiently covered in the paper and it is likely to explain the drop in performance on complex or non-rigid objects.
* Dependence on accurate segmentation, as it is based on reliable object masks (obtained via SAM2). In real world settings, due to lightning conditions and other occlusion factors, this could not be the case.
* The Object-centric representation (DetPool) is not really new. It could be useful to add (and compare with) two recent papers: https://arxiv.org/abs/2505.11563 and https://arxiv.org/abs/2503.11565.

**Questions:**

In the experimental section, you are often referring to a pre-defined grid when testing objects multiple times. Could you give more information about that pre-defined grid?

---

> ### Author Response · Authors · 2025-11-22
> **Response to Reviewer V8CY, Part 1**
>
> We thank the reviewer for their comments, and address the concerns raised below.
>
> **1. The approach is heuristic and does not support toy-based generalization:** We clarify that while the analogy with child development serves as the inspiration for our training data design, **our core focus is to demonstrate that object-centric representations induced by detection pooling are the key driver of object-level generalization**. To test this systematically, we constructed a rigorous experimental setup: training only on simple compositions of geometric primitives and evaluating zero-shot on real-world objects.
>
> We are **encouraged that other reviewers recognized this experimental design as robust and principled**. Reviewer `7VRd` noted that our results demonstrate how *"random combinations of a few geometric primitives suffice to support generalization"*, and Reviewer `oAEF` highlighted that *"training only on random compositions... is a nice way to study object-generalization"*. These assessments validate that our toy-based setup serves as an effective testbed for measuring generalization. Furthermore, our ablations rigorously confirm that this performance is not accidental: baselines using mean or attention pooling (which lack explicit object focus) perform substantially worse, proving that the object-centric structure is indeed the mechanism yielding generalizable features.
>
> Regarding physical properties, we fully agree that factors like mass, friction, and texture are critical for manipulation. However, we view visual/geometric generalization as a first step to physical interaction. Our work focuses on enabling zero-shot visual generalization for real world objects; extending the policy to account for physical properties (e.g., via learned physical latent vectors) is a complementary and promising direction for future research.
>
> **2. Dependence on accurate segmentation:** As the reviewer suggested, to quantitatively evaluate the dependence on accurate segmentation, we add a mask noise experiment.  Specifically, we **add randomized offset/noise to the bounding box** that is input to SAM for segmentation tracking. The noise is quantified in terms of Intersection over Union (IoU) of the ground truth bounding box and the noisy bounding box. The results are presented below.
>
> | IoU        | 250 demos | 2500 demos |
> |------------|-----------|------------|
> | **100% IoU**  | **56.63**     | **80.00**      |
> | **80% IoU**    | 55.83     | 79.29      |
> | **60% IoU**    | 53.19     | 77.98      |
>
> The results **indicate that model performance is robust to mask noise**. One reason for this level of robustness is that detection pooling sets the attention mask at the coarse patch level. Therefore, there is an inherent robustness to noise in the masks, as small inaccuracies are likely to result in the same patches still being chosen as object patches.
>
> In addition, we **add experiments to study the effect of factors such as lighting changes, clutter, and lookalike distractors** on the performance of our model. Specifically, we evaluate our model using 60% and 40% of the original lighting. For studying the effect of clutter, we add a random set of 2-5 YCB objects to the tabletop scene, in addition to the target object. Finally, for the lookalike distractor, we place another instance of the target object on the table.  The results are presented below.
>
>
> | Lighting       | 250 demos | 2500 demos |
> |----------------|-----------|------------|
> | **100% Lighting**  | **56.63**     | **80.00**      |
> | **70% Lighting**  | 56.44     | 79.33      |
> | **40% Lighting**   | 55.00     | 78.65      |
>
> | Condition   | 250 demos | 2500 demos |
> |-------------|-----------|------------|
> | **No Clutter** | **56.63**     | **80.00**      |
> | **Clutter**     | 46.73     | 67.21      |
>
> | Condition                    | 250 demos | 2500 demos |
> |-----------------------------|-----------|------------|
> | **No Distractor**              | **56.63**     | **80.00**      |
> | **Single Random Distractor**    | 50.19     | 71.35      |
> | **Single Lookalike Distractor** | 49.23     | 74.13      |
>
>
> Overall, **LEGO demonstrates robustness to lighting changes, clutter, and lookalike distractors**. This validates the use of the detection pooling which encourages greater focus on the target object region than other parts of the scene.

---

> ### Author Response · Authors · 2025-11-22
> **Response to Reviewer V8CY, Part 2**
>
> **3. Comparison with other object-centric approaches:** As suggested by the reviewer, we provide comparisons to the recommended works (Chapin et al. 2025, Emukpere et al. 2025) as well as to additional object-centric approaches  Zhu et al. 2023, Zhong et al. 2025). We are actively attempting to reproduce the suggested works; however, because they are not open-source, direct experimental comparison is non-trivial. We will continue these efforts and plan to include results, if successful, in the final version of the paper.
>
> We also **include GROOT and DexGraspVLA – both mask-based object-centric approaches – as experimental baselines in our YCB evaluation**, with results presented in the table below. While DexGraspVLA achieves moderate performance, it remains substantially outperformed by our detection pooling approach.
>
> | Method | 250 demos |2500 demos |
> |-|-|-|
> | **GROOT**  (Zhu et al., 2023)  | 0.87  | 1.54    |
> | **OpenVLA-OFT** (Kim et al., 2025) | 30.10              | 12.79               |
> | **$\pi_0$-FAST** (Black et al., 2024)| 8.85               | 4.13                |
> | **DexGraspVLA**  (Zhong et al., 2025)                   | 20.77                | 48.85                 |
> | **Ours**              | **56.63**              | **80.00**              |
>
>
> We qualitatively compare our approach to GROOT, DexGraspVLA, and the suggested works (Chapin et al. 2025, Emukpere et al. 2025) below, and **plan to add the comparisons below to the final manuscript.**
>
> - **Object-Centric Representations Improve Policy Generalization in Robot Manipulation (Chapin et al. 2025):** This work evaluates a variety of visual encoders, demonstrating that object-centric representations significantly enhance generalization, particularly under visual distribution shifts. These findings further validate the premise of our detection pooling mechanism: by focusing the model on grasping-relevant features, we enable effective zero-shot transfer to novel objects
>
> - **DOCIR (Disentangled Object-Centric Image Representation for Robotic Manipulation, Emukpere et al. 2025):** This work demonstrates that decomposing the scene into separate visual representations improves generalization. Specifically, it uses segmentation masks to generate and individually encode distinct images for the target object, robot, and obstacles. While LEGO shares the use of masks to focus on the target, it differs architecturally: rather than processing separate image inputs, LEGO uses the mask to constrain attention within the vision encoder. Experimentally, our policy demonstrates substantially stronger generalization; while DOCIR evaluates on just two object categories, LEGO successfully transfers from randomized toys to a wide variety of diverse real-world objects.
>
> - **GROOT (Learning Generalizable Manipulation Policies with Object-Centric 3D Representations, Zhu et al., 2023):** GROOT is an object-centric approach that predicts robot actions from object point clouds. Like LEGO, it tracks task-relevant objects using masks; however, while GROOT relies on backprojected 3D point clouds, LEGO uses masks to constrain attention in the vision encoder. Although point clouds enable generalization to structurally similar objects, they often struggle with completely new shapes, as shown by the results in the table below. In contrast, LEGO demonstrates stronger generalization—transferring directly from randomized toys to real-world objects—driven by the object-centric representation from detection pooling. We include GROOT as a baseline in our ManiSkill YCB benchmark below.
>
> - **DexGraspVLA (Zhong et al., 2025)** is a VLA model employing a planner-controller framework. It utilizes a large pre-trained VLM (Qwen2.5-VL-72B-Instruct) to decompose prompts into bounding-box instructions, which guide a diffusion controller. While sharing our object-centric reliance on target masks, key differences distinguish our approaches. DexGraspVLA depends on a massive VLM with internet-scale knowledge for high-level planning. In contrast, our model is orders of magnitude smaller, focusing specifically on grasping to demonstrate that the visual representation induced by detection pooling drives object-level generalization. We demonstrate that our approach offers a strong inductive bias that promotes learning efficiency, and view integrating detection pooling with pre-trained VLMs as a promising direction to extend this capability to broader manipulation tasks.
>
>
> **4. Evaluation grid:** To evaluate policies in simulation, we defined a 0.15 × 0.15 m square workspace, subdivided into a 4 × 4 grid. The grid was constructed from the Cartesian product of the sets {−0.075, −0.025, 0.025, 0.075} along both the x and y axes, resulting in 16 evenly spaced placements. For each trial, the object was placed at one grid location with its z-rotation initialized using a random seed. Each object was tested across all 16 placements, and success rates were averaged across objects and placements.

---

### Official Review · Reviewer_7VRd · 2025-10-31

**Soundness:** 3
**Presentation:** 4
**Contribution:** 3
**Rating:** 6
**Confidence:** 4

**Summary:**

This paper studies “cross-object generalization in grasp learning.” Inspired by developmental psychology, the authors train a grasping policy using only four geometric primitives (sphere, cube/rectangular prism, cylinder, and ring) that are randomly combined into “toys,” and achieve zero-shot generalization to real, unseen objects. The core idea is a “detection pooling” method called **DetPool** to build a **target-centric** visual representation: during visual encoding, a target segmentation mask constrains attention so object patches do not attend to background patches; the model then pools over the object patches to obtain a compact representation. A policy network takes this representation together with proprioception and predicts an action sequence (the overall framework is dubbed **LEGO**). Experiments show that training on only a few hundred random “toys” with limited demonstrations attains strong zero-shot grasp success on the real YCB object set. Compared to common pooling baselines (CLS token, mean pooling, attention pooling), DetPool yields markedly better generalization and scales well as data increases. Against several large vision-language-action baselines, LEGO is more robust under small-data and domain-shift settings; it also achieves leading or competitive success rates on a real Franka platform and a humanoid dexterous-hand platform, supporting the “target-centric representation + few demonstrations” path as a data-efficient solution.

Main contributions:

1. Proposes a “toy-driven” grasp learning paradigm and the **LEGO** framework, showing that random combinations of a few geometric primitives suffice to support generalization to complex real-world objects.
2. Introduces **DetPool**, which explicitly constructs a target-centric representation during visual encoding, significantly improving zero-shot and out-of-distribution generalization.
3. Provides systematic evaluation and scaling analyses in both simulation and real robots (including Franka and a dexterous-hand platform), demonstrating consistent gains across data scale, object diversity, and model size.

**Strengths:**

Originality
1.Proposes a “toy-driven” paradigm that composes a few geometric primitives to approximate real-world shape complexity, avoiding reliance on large annotated object corpora.
2.Introduces DetPool to build a target-centric representation during visual encoding (mask-constrained attention + pooling), offering a simple, effective object-centric alternative to CLS/mean/attention pooling.
Quality
Comprehensive evaluation: sim-to-real comparisons, strong VLA baselines, and ablations/scaling on pooling, data scale/diversity, and model size.
Clarity
 Coherent structure from motivation to analysis, clearly explaining why target-centric encoding and primitive-based training support generalization.
Significance
1. Demonstrates zero-shot grasping on complex real objects without heavy real-data/annotation costs, improving practical feasibility.
2.Provides concrete evidence for object-centric representations in embodied manipulation, nudging the field toward “structural priors + target-centric encoding.”
3. Cross-embodiment results suggest portability across manipulators/dexterous hands/mobile settings, indicating broad impact.

**Weaknesses:**

1. **Mask dependence and insufficient robustness**
   *Current setup:* GT masks in simulation, SAM2 in the real system; this may inflate DetPool’s gains.
   *Specific gaps:* No sensitivity curves of “mask quality → performance,” and no weak/no-prompt controls (box-only, no-mask).
   **Suggestion:** Use automatic (noisy) masks in both sim and real; plot performance vs IoU corruption, under/over-segmentation, and occlusion levels; add box-only and no-mask variants to quantify DetPool’s intrinsic benefit.

2. **Evaluation too idealized; extrapolation radius unclear**
   *Current setup:* Single-object, regular poses; the dexterous-hand setting uses only 13 objects with ~50% average success and high variance.
   *Specific gaps:* Missing stratified stress tests for clutter, multi-object scenes, heavy occlusion, look-alike distractors, with attribute-bucketed results.
   **Suggestion:** Build layered benchmarks and report attribute-bucketed success and failure attributions; provide “shift strength (viewpoint/material/scale) → success” curves to characterize the effective operating radius.

3. **Insufficient ablations; key claims are questionable**
   *Claim:* DetPool is the key to generalization and outperforms mean/attn/CLS pooling by 22–48%.
   *Specific gaps:*

   * All pooling variants share the same architecture/training recipe, but there’s no evidence they aren’t simply overfitting under limited data.
   * Missing **no-pooling** baseline using full-image features (full-image ViT).
   * No analysis of **SAM2** mask quality on performance (robustness to mask noise unclear).
   * SOTA baselines (e.g., OpenVLA-OFT 7B) overfit in small-data regimes; the paper doesn’t show LEGO remains superior at larger scales.
     **Suggestion:** Add data-regime sweeps (few→many), include a full-image no-pooling baseline, conduct mask-quality sensitivity analyses, and report matched-compute/parameter learning curves to verify whether DetPool’s advantage persists with more data.

**Questions:**

1. Authors use GT masks in simulation and SAM2 masks in the real setup. Could this amplify DetPool’s gains? Please unify to automatic (noisy) masks in both sim and real, report performance as a function of mask quality (e.g., IoU corruption, under/over-segmentation, occlusion levels), and include “box-only” and “no-mask” variants to quantify DetPool’s pure benefit under weak or absent prompts.
2. How is DetPool materially different from existing mask/ROI-guided object-centric representations (e.g., ROI pooling, mask-guided token selection, segmentation feature fusion)? Please compare against these closest alternatives under a fair hyperparameter grid and explain where DetPool’s advantage comes from (attention constraint vs pooling choice).
3. Your policy uses history length C=16 and predicts K=16 future actions. Please provide modular ablations where you hold the controller fixed and swap the visual representation (and vice versa). Also include latency/frame-rate vs success curves to show the gains are due to DetPool rather than increased temporal capacity or compute.
4.On Franka you use ~250 toys and ~1,500 successful demos; on the dexterous hand ~500 demos are needed. Please provide real-world scaling plots (toys count × diversity × demos → success), identify the “minimal viable toy set” and demo thresholds.
5. Quantify geometric similarity between training toys and target objects (e.g., Chamfer/SDF/voxel features), bucket by similarity, and report success with error bars. When targets diverge strongly from the primitives, does DetPool still generalize? Diagnose failure sources (grasp candidate generation, gripper aperture, approach strategy, etc.).
6.When mask boundaries are biased, have holes, or stick to background, how distorted is the representation? Do you need boundary refinement or multi-scale token aggregation to stabilize features? Please provide sensitivity analyses and any corrective post-processing that helps.

If the authors can address these questions convincingly during rebuttal, I am inclined to raise my score.

---

> ### Author Response · Authors · 2025-11-22
> **Response to Reviewer 7VRd, Part 1**
>
> We thank the reviewer for their insightful comments, and address the raised concerns below.
>
> **1. Mask dependence and insufficient robustness:**
>
> - **Mask noise.** As suggested, we conducted a quantitative experiment to evaluate the model's robustness to mask noise. Specifically, we evaluate our model by **adding randomized offset/noise to the bounding box** that is input to SAM for segmentation tracking. The noise is quantified in terms of Intersection over Union (IoU) of the ground truth bounding box and the noisy bounding box. The results are presented below, indicating that the model is robust to mask noise.
>
> | IoU        | 250 demos | 2500 demos |
> |------------|-----------|------------|
> | **100% IoU**  | **56.63**     | **80.00**      |
> | **80% IoU**    | 55.83     | 79.29      |
> | **60% IoU**    | 53.19     | 77.98      |
>
> We believe **our approach has inherent robustness to noise in the masks**, as small inaccuracies will almost always result in the same patches still being chosen as object patches since detection pooling sets the attention mask at the patch level.
>
> - **No-Mask & Box-only.** To quantify the intrinsic benefit of DetPool, we **evaluate two additional variants: box-only and no-mask**. The box-only variant provides the policy with a cropped image region containing the target object as input.  This baseline achieves a 0% success rate, which is to be expected since it does not allow the model to see where the target object is relative to the robot and the tabletop scene. The no-mask variant is equivalent to the mean pooling baseline already presented in Table 2 of the main paper, where the policy processes the entire scene without object-specific guidance.
>
> **2. Evaluation too idealized:** As suggested by the reviewer, we **add an experiment to study the robustness of our model in response to factors such as clutter, lookalike distractors, and lighting changes**. We present the results below.
>
> Specifically, we evaluate our model using 70% and 40% of the original lighting. For studying the effect of clutter, we add a random set of 2-5 YCB objects to the tabletop scene, in addition to the target object. Finally, for the lookalike distractor, we place another instance of the target object on the table.  The results are presented below.
>
> | Lighting       | 250 demos | 2500 demos |
> |----------------|-----------|------------|
> | **100% Lighting**  | **56.63**     | **80.00**      |
> | **70% Lighting**  | 56.44     | 79.33      |
> | **40% Lighting**   | 55.00     | 78.65      |
>
> | Condition   | 250 demos | 2500 demos |
> |-------------|-----------|------------|
> | **No Clutter** | **56.63**     | **80.00**      |
> | **Clutter**     | 46.73     | 67.21      |
>
> | Condition                    | 250 demos | 2500 demos |
> |-----------------------------|-----------|------------|
> | **No Distractor**              | **56.63**     | **80.00**      |
> | **Single Random Distractor**    | 50.19     | 71.35      |
> | **Single Lookalike Distractor** | 49.23     | 74.13      |
>
> Overall, **LEGO demonstrates robustness to lighting changes, clutter, and lookalike distractors**, due to the detection pooling which encourages greater focus on the target object region than other parts of the scene.

---

> ### Author Response · Authors · 2025-11-22
> **Response to Reviewer 7VRd, Part 2**
>
> **3. Insufficient ablations; key claims are questionable:**
>
> - **Pooling Variants Overfitting:** We clarify that Table 1 reflects a zero-shot evaluation: models are trained on toys and evaluated on unseen real-world objects. This domain gap makes overfitting to the evaluation set unlikely. Additionally, to ensure fair comparisons, **all baselines were tuned across a comprehensive hyperparameter grid**, and we report the best performance for each.
>
> - **Full-image, no-pooling baseline:** A full-image, no-pooling baseline would require the policy to process 257 tokens per image (256 spatial tokens plus 1 CLS token for patch size=16). For any reasonable context length, this represents a prohibitively large number of image tokens to feed into the policy, a problem exacerbated by adding multiple views and thus making such an approach impractical. In contrast, pooling methods such as CLS and mean pooling allow the model to preserve information from the full image efficiently, reducing the number of tokens while retaining a comprehensive representation.
>
> - **SAM mask quality:** Please refer to answer point 1.
>
> - **SOTA baseline performance:** The state-of-the-art baselines we tested are designed to be fine-tuned on new settings, leveraging their large-scale pretraining. As shown in Table 2 of the main paper, **increasing the data pool from 250 to 2500 demonstrations does not significantly improve performance for these baselines**. This poor performance is instead likely due to our zero-shot evaluation setting, with both baselines struggling to handle novel objects not seen during training. Importantly, **data efficiency is a key strength of our approach**: instead of training on a large set of real-world objects, we train on a small set of randomized toys and achieve zero-shot generalization to new objects.
>
> **4. DetPool materially different from existing mask/ROI-guided object-centric representations:** As suggested by the reviewer, we **compare our method to related mask-guided object-centric approaches in robotic manipulation**, and will also add these comparisons in the final version of the paper. Specifically, we add GROOT (Learning Generalizable Manipulation Policies with Object-Centric 3D Representations) and DexGraspVLA as experimental baselines, with the results presented below.
>
> | Method                | 250 demos |2500 demos |
> |----|-----|----|
> | **GROOT**  (Zhu et al., 2023)  | 0.87                 | 1.54                 |
> | **OpenVLA-OFT** (Kim et al., 2025) | 30.10              | 12.79               |
> | **$\pi_0$-FAST** (Black et al., 2024)| 8.85               | 4.13                |
> | **DexGraspVLA**  (Zhong et al., 2025)                   | 20.77                | 48.85                 |
> | **Ours**              | **56.63**              | **80.00**              |
>
>
> The first new baseline, GROOT, uses object-centric point clouds derived from segmentation masks, to predict robot actions. This approach can be difficult to scale and deploy in the real world, and as seen by the results, also does not generalize well from toys to real-world objects. Even with the use of ground-truth depth data from the simulator, this approach is unable to adapt when encountering novel objects. The other object-centric baseline DexGraspVLA incorporates segmentation masks directly in the visual representation. This baseline achieves decent performance, getting higher success rates than OpenVLA-OFT and $\pi_0$-FAST. However, DexGraspVLA **still does not reach the performance levels unlocked by detection pooling**, and its performance does not scale with increasing amounts of data as well.

---

> ### Author Response · Authors · 2025-11-22
> **Response to Reviewer 7VRd, Part 3**
>
> **5. Ablations on condition and prediction length:** As suggested by the reviewer, we conduct an experiment that studies the effect of changing condition length $c$ and prediction length $k$ on the performance of the model. We present the results below.
>
> | $c$ (Condition Length) | $k$ (Prediction Length) | 250   | 500   | 1000  | 1500  | 2000  | 2500  |
> |------------------------|------------------------|-------|-------|-------|-------|-------|-------|
> | **16**                     | **4**                      | 63.65 | 71.92 | 63.94 | 69.85 | 69.42 | 70.67 |
> |  **16**                      | **8**                      | 59.04 | 69.90 | 68.85 | 77.31 | 73.27 | 78.65    |
> |  **16**                      |  **16**                      | **56.63** | **68.17** | **71.15** | **74.62** | **76.83** | **80.00** |
> | **8**                      |  **16**                      | 38.94 | 54.71 | 57.60 | 62.31 | 70.00 | 71.15 |
> | **4**                      |  **16**                      | 22.12 | 30.38 | 38.46 | 52.07    | 52.88   | 54.30   |
>
> From the bottom three rows, it is evident that a lower condition length has a substantial negative impact on performance. In contrast, the top three rows show that prediction length is comparatively less important, although longer prediction lengths still improve performance.
>
> **6. Geometric similarity between training toys and target objects:** As suggested by the reviewer, we analyzed the geometric similarity between the training toys and the target objects. Specifically, we computed the Chamfer distance between each target object and its closest counterpart in the training set. The target objects were then grouped into “easy,” “medium,” and “hard” categories based on the 33rd and 66th percentiles of the Chamfer distances. For each difficulty bucket, we report the average success rate along with a 95% confidence interval, as well as summary statistics of the Chamfer distances.
>
> | Difficulty | Num. of Objects | Success  | CD (Chamfer Distance) Mean | CD Std | CD Min | CD Max |
> |-------------------|------------|-------------------|------------------|-----------------|-----------------|-----------------|
> | **Easy**              | **22**         | $75.6 \pm 13.8$ | 0.0568           | 0.0123          | 0.0392          | 0.0756          |
> | **Medium**            | **21**         | $83.0 \pm 12.0$ | 0.0990           | 0.0070          | 0.0871          | 0.1080          |
> | **Hard**              | **22**         | $81.5 \pm 12.7$ | 0.1300           | 0.0283          | 0.1082          | 0.2164          |
>
>  These results indicate that while the Chamfer distance increases across buckets, the average success rate remains relatively consistent. This suggests **that the model is robust to geometric variation in the target objects**, and that the difference in similarity to training toys does not significantly impact grasp success, demonstrating strong object-level generalization.

---

> ### Comment · Reviewer_7VRd · 2025-11-27
> **Official Comment by Reviewer 7VRd**
>
> I thank the authors for the extensive additional experiments. While these address many of my concerns, some questions remain incompletely resolved. I maintain my score of 6 and defer to the AC and fellow reviewers for the final decision.

---

> > ### Author Response · Authors · 2025-11-27
> >
> > We thank the reviewer for the follow-up and for acknowledging that the additional experiments address many of the concerns. We would appreciate clarification on any points that remain incompletely resolved, and we will provide targeted responses to fully resolve the reviewer’s remaining concerns.

---

### Official Review · Reviewer_oJDK · 2025-11-01

**Soundness:** 3
**Presentation:** 3
**Contribution:** 2
**Rating:** 6
**Confidence:** 4

**Summary:**

This paper presents a method LEGO (LEarning to Grasp from tOys), to grasp diverse objects by learning to grasp random, synthetically generated toys composed of primitive 3D geometries (spheres, cuboids, cylinders, and rings). For each toy, a random number of primitives of random sizes are fused in random positions and orientations. In total, 250 such toys are generated in simulation as well as 3D printed in the real world. Grasping data is primarily collected via teleoperation except for single primitives, which are collected with motion planning.

A neural network is trained with behavior cloning on the grasp demonstrations. The architecture consists of a Vision Transformer (ViT) for extracting visual features. Authors use a pretrained MVP. SAM 2 is used to get a segmentation mask of the object of interest. Attention is masked between object and background patches using the segmentation mask. Finally, features in object patches are mean-pooled to get a vision feature. This masking mechanism and pooling is coined Detection Pooling (DetPool).

The vision feature is concatenated with proprioception to get a single token at a time step. Multiple tokens from the history of timesteps are passed to a transformer policy to predict the next action token.

Results, both in simulation and real robot, show competitive performance, even when compared to the state-of-the-art VLAs.

**Strengths:**

**Positive result**
The authors show results in simulation as well as the real world on 2 robot platforms (Franka robot and Unitree H1-2), comparing their method against state-of-the-art VLAs (both scratch and finetuned). In simulation and Unitree H1-2, the proposed method outperforms all baselines. In real Franka, the proposed method outperforms all baselines except finetuned $\pi_0-FAST$

**Extensive ablations**
1. As they increase no. of toys, performance increases initially but quickly saturates with no/minila difference beyond 25 toys
2. As they increase no. of demonstrations, performance increases
3. As they increase model size, performance improves and then saturates. 86M is the sweet spot for their setting
4. Sphere is the most important primitive shape as removing it leads to significant drop in performance. Ring has the least impact
5. Toy complexity (defined by no. of primitives): 2 primitives contributes the most. Likely due to the level of complexity of toys in the test set

**Weaknesses:**

**Doesn't capture desired grasp location**
Many objects have desired grasp location, e.g. a cup should be grapsed by it's handle & a vertical cylindrical object is more stably grasped sideways instead of top-down. The proposed method finds a grasp location but doesn't guarantee grasp at desired grasp location since grasping is learnt on random toys.

**Doesn't capture physical properties of objects**
Some objects may have delicate parts necessitating that the robot doesn't apply too much force which grasping. As such, it's a stretch to claim that the method can grasp any object as per the title. A better wording would be - diverse real world objects or something similar.

**Constraints imposed by masking**
This makes it more difficult to extend to complex manipulation tasks involving multiple objects or occlusion

**Minor comments**
1. Line 191: Grammar - "details our including preliminaries"
2. Line 197: Definining visual observations as $i_{1:T}$ is incorrect since at $t=1$, visual observation is $i_{1-C+1:1}$ which includes time $t<1$ which is outside the $1:T$ range. Suggestion - We denote visual observation at time $t$ as $i_t$ and proprioceptive state as $s_t$. Observation at time $t$ consists of $i_{t-C+1:t}, s_{t-C+1:t}$.
3. Line 425: "Effect of Number and Diversity of Demonstrations" - Diversity can come from grasp style variation and not just object variation. Suggestion - Effect of Number Demonstration and Number of unique toys
4. Table 4 and 5: Label the entries. No. of demonstration and success rate.
5. In conclusion, it's not entirely correct to claim that the method outperforms baseline VLAs since for Franka finetuned $\pi_0 FAST$ significantly outperforms.
6. Line 403: Missing citation of ManiSkill
7. Line 425: Missing details on test objects, no. of evals, etc

**Questions:**

1. What was the reason for choosing absolute instead of delta joint prediction?
2. Line 307: "our unique 250 toys". Line 341: "We 3D print the 250 toys with the highest". If 250 is the total no. of toys, how many of them were selected for Franka experiment?
3. What is the performance trend when trained between 1 and 25 toys?
4. What were the primary failure modes? What's preventing higher success rates, say 90%, 99% or even 100%?

---

> ### Author Response · Authors · 2025-11-22
> **Response to Reviewer oJDK, Part 1**
>
> We thank the reviewer for the insightful comments, and address each concern below.
>
> **1. Performance trend between 1 and 25 toys:** As suggested by the reviewer, we add two additional toy sets to expand our ablation study on the number of toys. Specifically, we train our model on sets with 5 and 15 unique toys respectively. We present the results below.
>
> | Toys        | 25    | 125   | 250   | 500   | 1000  | 1500  | 2000  | 2500  |
> |-------------|-------|-------|-------|-------|-------|-------|-------|-------|
> | **1 toy**   | 1.83  | 4.33  | 8.85  | 14.52 | 24.62 | 24.71 | 25.00 | 26.35 |
> | **5 toys**  | 4.04  | 10.38 | 24.52 | 34.71 | 46.63 | 51.54 | 56.06 | 58.27 |
> | **15 toys** | 15.58 | 26.92 | 45.10 | 58.46 | 61.92 | 69.90 | 70.58 | 72.40 |
> | **25 toys** | 18.37 | 39.42 | 49.52 | 61.35 | 69.23 | 71.92 | 76.25 | 76.63 |
>
>
> It can be seen that the **new data points of 5 and 15 unique toys follow the trend previously established**, with an increasing number of demos improving performance.
>
> **2. Absolute instead of delta joint prediction:** We chose absolute joint control rather than delta control in all our experiments since we experimentally found it to perform better. We present a comparison of the two control modes below, across a varying number of demonstrations for a more comprehensive analysis.
>
> | Method           | 250   | 500   | 1000  | 1500  | 2000  | 2500  |
> |------------------|-------|-------|-------|-------|-------|-------|
> | **Delta Control**    | 29.71 | 30.19 | 31.63 | 34.90 | 36.44 | 37.02 |
> | **Absolute Control** | **56.63** | **68.17** | **71.15** | **74.62** | **76.83** | **80.00** |
>
> It can be seen that **absolute control performs much better, validating our choice of control mode**.
>
> **3. Doesn't capture desired grasp location:** As the reviewer noted, our method does not explicitly account for semantic affordance (e.g., grasping a handle). This is an inherent characteristic of training on randomized toys, which lack semantic labels. We emphasize, however, that **our primary focus in this work was to demonstrate that learning from these toys enables zero-shot object-level generalization**. To address desired grasp locations, we view extending our approach with affordance heatmaps or real-world fine-tuning as a promising direction for future work.
>
> **4. Doesn't capture physical properties of objects:** We agree with the reviewer that physical properties such as mass distribution, friction, and texture play a critical role in manipulation. We believe that incorporating these object characteristics into the policy—for instance, via a learned physical latent vector—would further enhance performance. While our current work provides a first step toward real-world generalization from toy objects using geometric primitives, we **view the integration of physical dynamics as a promising direction for future research**. Finally, we agree with the comment regarding the title and will revise it in the final version.

---

> ### Author Response · Authors · 2025-11-22
> **Response to Reviewer oJDK, Part 2**
>
> **5. Constraints imposed by masking:** To study the robustness of our model, we conduct experiments examining the **effect of adding clutter and multiple objects to the scene**. The results are presented below.
>
> | Condition   | 250 demos | 2500 demos |
> |--|---|--|
> | **No Clutter** | **56.63**     | **80.00**      |
> | **Clutter**     | 46.73     | 67.21      |
>
> | Condition                    | 250 demos | 2500 demos |
> |-|--|-|
> | **No Distractor**              | **56.63**     | **80.00**      |
> | **Single Random Distractor**    | 50.19     | 71.35      |
> | **Single Lookalike Distractor** | 49.23     | 74.13      |
>
> LEGO **remains robust to clutter and lookalike distractors** due to the detection pooling which encourages greater focus on the target object region than other parts of the scene.
>
>
> We further note that **the masks input to our model do not need to be highly precise**. Detection pooling operates at the patch level and can tolerate noise in the masks, making the model inherently robust to minor inaccuracies.
> Specifically, we evaluate our model by adding randomized offset/noise to the bounding box that is input to SAM for segmentation tracking. The noise is quantified in terms of Intersection over Union (IoU) of the ground truth bounding box and the noisy bounding box. The results are presented below, indicating that **the model is robust to mask noise**.
>
> | IoU        | 250 demos | 2500 demos |
> |------------|-----------|------------|
> | **100% IoU**  | **56.63**     | **80.00**      |
> | **80% IoU**    | 55.83     | 79.29      |
> | **60% IoU**    | 53.19     | 77.98      |
>
> **6. Minor comments:** Thank you, we will fix these in the final version of the paper.
>
> **7. Number of toys in set:** As described in our ablation studies in Sec 5.5, we conducted a simulation experiment to determine the optimal number of training toys. We found that 250 toys maximized performance; therefore, we utilized this set for all simulation and real-world experiments, ensuring consistent comparisons with baselines.
>
>
> **8. Primary failure modes:** The primary failure mode of our model is imprecise grasping. Given that the model generalizes zero-shot from randomized toys to real-world objects, it occasionally struggles to form stable grasps on highly novel structures due to the domain gap. We believe that improvements, as suggested by the reviewer, such as modeling physical object properties and grasping locations, can help make the grasping results even more robust.

---

### Official Review · Reviewer_oAEF · 2025-11-01

**Soundness:** 3
**Presentation:** 3
**Contribution:** 2
**Rating:** 4
**Confidence:** 4

**Summary:**

The paper proposes training robotic grasping policies exclusively on randomly assembled “toys” made from four shape primitives, then evaluating zero-shot on real objects. A detection pooling mechanism (DetPool) is introduced to induce object-centric visual representations. Experiments in simulation and on two real robot platforms (a Franka with a parallel gripper and a humanoid with dexterous hands) show strong transfer, including notable gains over large pretrained vision-language-action models under several settings. Ablations study the role of toy diversity, number of demonstrations, model size, and primitive/complexity composition.

**Strengths:**

- Clear and compelling motivation with an interesting set of experiments
- Training only on random compositions of four primitives is a nice way to study object-generalization.
- The procedure for generating and 3D-printing toys is well specified and reproducible.
- Simple, modular representation idea with strong empirical effect:
- Results in simulation and a real Franka setup on 64 YCB objects, and a humanoid with dexterous hands
- The study disentangles the effects of number of unique toys versus demonstrations per toy, showing stronger gains from more demonstrations once a small toy set is present (Figure 4, left). This is a useful, actionable finding for data collection planning.
- Detailed reporting of implementation and experiment setup in the appendix

**Weaknesses:**

- Inconsistency/ambiguity in DetPool’s implementation and mechanism:
  - Section 4.3 states that the object mask is used to enforce no attention between object and non-object tokens inside the vision transformer and then mean-pool object tokens. Appendix E, however, describes patchifying, passing through transformer blocks, and then using the mask only at pooling time to select spatial features. This distinction matters: masked self-attention changes the encoder’s computation; masked pooling does not. The paper should explicitly reconcile which variant is used in each experiment (simulation vs. real) and in Table 1’s ablations, and whether both variants were tried and compared head-to-head.
- Mask generation pipeline is split across sections, creating avoidable ambiguity:
  - Section 5.1 states SAM2 for real and ground-truth masks for simulation, while Appendix E explains the two-stage real pipeline (Faster R-CNN boxes → SAM2 masks). The pipeline is present, but unifying its description in the main text would reduce confusion and make the experimental setup easier to follow.
- Unfair evaluation of baselines:
  - LEGO has access to object masks (ground-truth in simulation, detector→SAM2 pipeline in real), whereas competing VLAs (OpenVLA-OFT, π0-FAST) are not given any object cues (e.g., crop/box/mask). Since the paper’s main contribution is about the value of object-centric features, this mismatch likely affects the performance gap. A fair comparison should provide comparable object-centric inputs (or report variants with/without such cues for all methods).
- Mixed messaging on “outperforming state-of-the-art”:
  - The abstract claims outperforming state-of-the-art approaches trained with much more in-domain data. In Table 2, a fine-tuned π0-FAST achieves higher real-world success than LEGO on the Franka setting. The paper discusses this, but the abstract/overall positioning would benefit from qualifying the claim or toning down these claims.
- Evaluation protocol mismatches and thresholds:
  - In simulation, the success threshold for OpenVLA is relaxed to 0.15 m (vs. 0.3 m for others) due to gripper reopening (Appendix H.1). Although documented, thresholds should be unified or the relaxation carefully justified and accompanied by sensitivity analysis.
  - Sensor and control mismatches exist across methods (e.g., camera views, absolute vs. delta joint control). While some of these are inherited from baseline codebases, the paper should quantify or mitigate their impact.
- Vision encoder training status unclear and parameter counts potentially misleading:
  - The paper uses a ViT-L MVP encoder (Section 5.1), but it is not explicitly stated whether it is frozen or fine-tuned in each setting. Reporting whether gradients flow to the encoder is important for reproducibility and for fair comparison to large VLAs that train all vision stacks.
- Missing or limited analyses that would strengthen the generalization claim:
  - Failure-mode and robustness analyses (lighting changes, heavy clutter, occlusion, distractors with similar colors/shapes, heavy texture) are limited. Given the object-centric focus, stress tests around segmentation failures (mask noise, false positives/negatives) would be highly informative.
  - An oracle-cropping or oracle-bounding-box control for baselines would isolate the contribution of precise object localization versus the transformer architecture itself.

**To the authors:** I find the motivation behind this work compelling and see potential for a strong contribution. I’d be glad to reconsider my evaluation if you can effectively address the issues highlighted in the review and outline a concrete plan to strengthen the experimental validation.

**Questions:**

1. Why only consider Transformer architectures here? It seems to me this could be applicable to CNNs as well?
2. Can you clarify precisely which variant of DetPool is used in your main experiments — masked self-attention plus pooling, or masked pooling only. Were they both compared head-to-head?
3. Since LEGO uses explicit object masks while other baselines do not, did you evaluate variants of the baselines with comparable object-localization inputs (e.g., crops or masks)? If not, how do you justify the fairness of these comparisons?
4. Is the ViT-L MVP encoder frozen or fine-tuned in each experiment? If partially fine-tuned, could you specify which layers receive gradients?
5. The model is described as having 86M parameters. Does this exclude the ViT-L encoder? Could you report both total and trainable parameter counts for transparency?
6. Why is the success threshold for OpenVLA relaxed relative to other methods? Did you perform sensitivity analyses to confirm that this does not alter comparative trends?
7. Some baselines use different control modes and camera setups. Can you quantify or comment on how much these differences might affect performance comparisons?
8. Have you tested how performance degrades under realistic mask noise, detection failures, or segmentation errors? Since DetPool relies on accurate masks, this seems crucial for generalization.
9. Could you include analyses under more challenging conditions, such as heavy clutter, occlusion, lighting variation, or distractor object, to better support claims of robust object generalization?
10. Have you compared DetPool against simpler alternatives such as crop-and-encode or mask-as-input-channel baselines (e.g., Shariar et al. 2025) to isolate the contribution of attention masking itself?
11. The abstract claims outperforming state-of-the-art approaches, though π0-FAST surpasses LEGO in one setup. Could you clarify or qualify which baselines and settings support this claim?


**References**
1. Shahriar, F., Wang, C., Azimi, A., Vasan, G., Elanwar, H. H., Mahmood, A. R., & Bellinger, C. (2025). General and Efficient Visual Goal-Conditioned Reinforcement Learning using Object-Agnostic Masks. arXiv preprint arXiv:2510.06277.

---

> ### Author Response · Authors · 2025-11-22
> **Response to Reviewer oAEF, Part 1**
>
> We thank the reviewer for their insightful comments, and address the concerns raised below.
>
> **1. Inconsistency/ambiguity in DetPool’s implementation and mechanism:** We thank the reviewer for highlighting this point. We confirm that all experiments utilized the detection pooling implementation detailed in Section 4.3: the object mask is used to enforce the Vision Transformer to prevent attention between object and non-object tokens, after which the object tokens are aggregated via mean-pooling. As the reviewer mentioned, the description in the Appendix E is inaccurate, and we will correct this in the final version of the manuscript.
>
> **2. Mask generation pipeline is split across sections, creating avoidable ambiguity:** Thank you for this suggestion to help improve clarity. We will fix this in the final version of the paper, unifying the description of the pipeline in the main text, as you suggested.
>
> **3. Unfair evaluation of baselines:** As noted by the reviewer, unlike our model, the VLA baselines used for comparison are not given any explicit object cues. **We add a version of these baselines by providing them with ground-truth bounding box information for the target object as part of the proprioception input**, and evaluate them on the ManiSkill YCB dataset. The results are presented below. It can be seen that the performance of OpenVLA worsens, indicating that object cues do not help improve performance, while the performance of $\pi_0$-FAST only marginally improves. This lack of significant performance improvement with the addition of object bounding boxes could potentially be because of the lack of such object-centric information during the large-scale pretraining for these models.
>
> | **Method**                 | 500 demos | 2500 demos |
> |---------------------------|-----------|------------|
> | **OpenVLA**               | 36.35     | 12.79      |
> | **OpenVLA (with bbox)**   | 12.88     | 11.15      |
> | **$\pi_0$-FAST**          | 7.60      | 4.13       |
> | **$\pi_0$-FAST (with bbox)** | 9.62  | 6.44       |
> | **Ours - Attn Pooling**   | 40.10     | 51.63      |
> | **Ours - Cls Pooling**    | 20.29     | 49.81      |
> | **Ours - Mean Pooling**   | 30.38     | 40.58      |
> | **Ours — Det Pooling**    | **68.17**     | **80**         |
>
>
> We additionally include **two object-centric baselines that explicitly incorporate object masks into their visual processing pipelines**: GROOT (Learning Generalizable Manipulation Policies with Object-Centric 3D Representations, Zhu et al., 2023) and DexGraspVLA (Zhong et al., 2025). GROOT leverages a point-cloud–based object-centric representation, while DexGraspVLA integrates object masks directly into its visual encoder. We train both methods on the set of 250 randomized toys and evaluate their zero-shot performance on the YCB benchmark. The results are reported below. It can be seen that our method outperforms both new methods by a significant margin.
>
> | Method                | 250 demos |2500 demos |
> |---------------------------------|--------------------|---------------------|
> | **GROOT**  (Zhu et al., 2023)                        | 0.87                 | 1.54                 |
> | **OpenVLA-OFT** (Kim et al., 2025) | 30.10              | 12.79               |
> | **$\pi_0$-FAST** (Black et al., 2024)| 8.85               | 4.13                |
> | **DexGraspVLA**  (Zhong et al., 2025)                   | 20.77                | 48.85                 |
> | **Ours**              | **56.63**              | **80.00**              |
>
>
> **4. Mixed messaging on “outperforming state-of-the-art”:** We thank the reviewer for the suggestion. To clarify, the abstract referred to the zero-shot version of $\pi_0$, which uses a checkpoint pretrained on the DROID Franka setup. However, we fully agree with the reviewer’s feedback regarding the framing. We will modify this claim in the final manuscript and tone down the language as suggested.

---

> ### Author Response · Authors · 2025-11-22
> **Response to Reviewer oAEF, Part 2**
>
> **5. Evaluation protocol mismatches and thresholds:**
> - **OpenVLA evaluation threshold:** We set the OpenVLA evaluation threshold to 0.15m to account for a consistent gripper-reopening behavior observed in simulation. This value was determined via a calibration study measuring the average lift height before premature release; using the standard threshold of 0.3m classified nearly all valid OpenVLA grasps as failures. Crucially, our policy outperforms OpenVLA despite being evaluated on the stricter standard metric (0.3m). **We further confirm that relaxing the threshold to 0.15m for our model yields identical results**, as our policy maintains a stable grasp and does not suffer from premature release.
> - **Sensor and control mismatches:** For all baselines, we used the same control settings as in their respective pretraining phases to preserve the benefits of their pretrained representations. For example, because OpenVLA is pretrained with delta end-effector control, we adopt the same setting in our experiments. Similarly, since $\pi_0$ is pretrained with joint-space control, we keep that setting unchanged. Since the pretraining dataset for $\pi_0$ is not publicly available, we ablated absolute versus delta joint–space control and ultimately adopted absolute control, which yielded better performance.
>
> **6. Vision encoder training status unclear and parameter counts potentially misleading:** To clarify, the vision encoder remains frozen in all our experiments. We apologize for the ambiguity and will explicitly state the breakdown in the final manuscript: The total number of parameters is 393M, while the number of trainable parameters of our model is 86M. We note that our original reporting aligns with standard conventions for VLAs/VLMs (e.g., OpenVLA - a 7B decoder), which typically cite the size of the trainable decoder while excluding the frozen vision encoder.
>
> **7. Effect of mask noise:** As suggested by the reviewer, we add an experiment to study the effect of mask noise on our model’s performance. Specifically, **we add randomized offset/noise to the bounding box** that is input to SAM for segmentation tracking. The noise is quantified in terms of Intersection over Union (IoU) of the ground truth bounding box and the noisy bounding box. The results are presented below.
>
> | IoU        | 250 demos | 2500 demos |
> |------------|-----------|------------|
> | **100% IoU**  | **56.63**     | **80.00**      |
> | **80% IoU**    | 55.83     | 79.29      |
> | **60% IoU**    | 53.19     | 77.98      |
>
> The results clearly indicate that **model performance is robust to mask noise**. One reason for this level of robustness is that detection pooling sets the attention mask at the patch level. Therefore, there is an inherent robustness to noise in the masks, as small inaccuracies are likely to result in the same patches still being chosen as object patches.

---

> ### Author Response · Authors · 2025-11-22
> **Response to Reviewer oAEF, Part 3**
>
> **8. Robustness analysis:** We **add experiments to study the effect of factors such as lighting changes, clutter, and lookalike distractors** on the performance of our model. Specifically, we evaluate our model using 70% and 40% of the original lighting. For studying the effect of clutter, we add a random set of 2-5 YCB objects to the tabletop scene, in addition to the target object. Finally, for the lookalike distractor, we place another instance of the target object on the table.  The results are presented below.
>
> | Lighting       | 250 demos | 2500 demos |
> |----------------|-----------|------------|
> | **100% Lighting**  | **56.63**     | **80.00**      |
> | **70% Lighting**  | 56.44     | 79.33      |
> | **40% Lighting**   | 55.00     | 78.65      |
>
> | Condition   | 250 demos | 2500 demos |
> |-------------|-----------|------------|
> | **No Clutter** | **56.63**     | **80.00**      |
> | **Clutter**     | 46.73     | 67.21      |
>
> | Condition                    | 250 demos | 2500 demos |
> |-----------------------------|-----------|------------|
> | **No Distractor**              | **56.63**     | **80.00**      |
> | **Single Random Distractor**    | 50.19     | 71.35      |
> | **Single Lookalike Distractor** | 49.23     | 74.13      |
>
> Overall, **LEGO demonstrates robustness to lighting changes, clutter, and lookalike distractors**, due to the detection pooling which encourages greater focus on the target object region than other parts of the scene.
>
> **9. Crop-box, mask-as-input-channel baselines:** As suggested by the reviewer, **we add a crop-box baseline that takes the cropped region of the target object as its visual input**. This baseline achieves a 0% success rate, which is to be expected since it does not allow the model to see where the target object is relative to the robot and the tabletop scene. In addition, we note that a mask-as-input-channel vision encoder as implemented in Shariar et. al. (2025) would require a vision encoder learned from scratch (to accommodate the extra channel). This fundamentally conflicts with our approach of leveraging large-scale pretrained vision encoders, whose performance advantage comes from having been trained on billions of RGB-only images.
>
> **10. CNN vs ViT as the vision encoder:** Detection pooling **could be applied to CNNs as well**. In this work, we use ViTs for the vision encoder due to their proven efficacy in the robot domain in several other foundational robotic models such as Octo, OpenVLA, and $\pi_0$. However, we do believe that detection pooling could be used effectively with a CNN vision encoder as well.

---

### Official Review · Reviewer_CYqi · 2025-11-02

**Soundness:** 2
**Presentation:** 2
**Contribution:** 2
**Rating:** 4
**Confidence:** 4

**Summary:**

This work introduces LEGO, demonstrating that robots can acquire robust general-purpose grasping skills by learning from a simple set of objects composed of just four basic shape primitives: spheres, cuboids, cylinders, and rings.

A key contribution of this paper is the use of a detection pooling mechanism to learn a critical object-centric visual representation, enabling the policy to generalize to a wide range of real-world objects in a zero-shot manner.

**Strengths:**

- The paper executes a significant number of real-world robot experiments, which is excellent.

- The idea of using randomly assembled objects that are composed from just four shape primitives—spheres, cuboids, cylinders, and rings is insightful.

**Weaknesses:**

- The detection pooling proposed in this paper shares similar ideas with those widely applied in object-centric manipulation. The authors should discuss these related works and clearly articulate the differences between them and their own approach. For instance, the following works:

[1] Learning Generalizable Manipulation Policies with Object-Centric 3D Representations

[2] Transferring foundation models for generalizable robotic manipulation

[3] DexGraspVLA: A Vision-Language-Action Framework Towards General Dexterous Grasping

- Another major shortcoming lies in the paper's evaluation. First, the paper only evaluates the grasp task. Evaluating other contact-rich manipulation tasks is necessary, as this is central to distinguishing the proposed method from pure grasping approaches like Anygrasp. Second, the comparison made against models such as FAST and OpenVLA-OFT is unfair, as the evaluation task is a single-task setup that does not require language understanding or reasoning.

**Questions:**

Please see the Weaknesses section.

---

> ### Author Response · Authors · 2025-11-22
> **Response to Reviewer CYqi, Part 1**
>
> We thank the reviewer for the insightful comments, and address the concerns raised below. We will add the comparisons to other works in the Related Work and Experiments sections in the final manuscript.
>
> **1. Quantitive Comparison with other Object-Centric Approaches:** As suggested by the reviewer, we compare our method to related object-centric approaches in robotic manipulation. Specifically, we add GROOT (Learning Generalizable Manipulation Policies with Object-Centric 3D Representations) and DexGraspVLA as experimental baselines in ManiSkill, with the results presented below.
>
> | Method                | 250 demos |2500 demos |
> |---------------------------------|--------------------|---------------------|
> | **GROOT**  (Zhu et al., 2023)                        | 0.87                 | 1.54                 |
> | **OpenVLA-OFT** (Kim et al., 2025) | 30.10              | 12.79               |
> | **$\pi_0$-FAST** (Black et al., 2024)| 8.85               | 4.13                |
> | **DexGraspVLA**  (Zhong et al., 2025)                   | 20.77                | 48.85                 |
> | **Ours**              | **56.63**              | **80.00**              |
>
> The first new baseline, GROOT, uses object-centric point clouds to predict robot actions. This approach can be difficult to scale and deploy in the real world, and as seen by the results, also does not generalize well from toys to real-world objects. Even with the use of ground-truth depth data from the simulator, this approach is unable to adapt when encountering novel objects. The other object-centric baseline DexGraspVLA achieves decent performance, getting higher success rates than OpenVLA-OFT and $\pi_0$ -FAST. However, it still does not reach the performance levels unlocked by detection pooling, and its performance does not scale with increasing amounts of data as well.
>
> **2. Evaluating on other tasks:** As the reviewer noted, our experiments evaluate the effectiveness of our detection pooling by focusing on the grasping task. We note that this work places specific emphasis on grasping, a simple yet significant manipulation task, in order to assess our object-centric approach. While we acknowledge this is a first step, we believe it is an important one toward achieving robust generalization across a wide range of real-world tasks. In addition, **our methodology is not tailored to the grasping task in any way, and can be used for other general manipulation tasks as well.**
>
> Following the reviewer’s suggestion, we have added an additional manipulation task, “push object to target”, which requires precise contact and force to slide an object to a target region, and is a very different manipulation task compared to grasping. As before, we train our model on the randomized set of 250 unique toys and evaluate it zero-shot on the YCB object benchmark. We present our results below, including OpenVLA-OFT as a baseline.
>
> | Method | 250 demos | 2500 demos |
> |-----------------|--------------------|---------------------|
> | **OpenVLA**         | 58.46                 | 75.77                |
> | **Ours**            | **67.31**                 | **81.83**        |
>
> From the results, it can be seen that the advantages of detection pooling persist in a different task. While OpenVLA-OFT improves on its performance compared to the grasping task, its success rate is still significantly lower than our method’s. In addition, the performance scales with the number of demonstrations, a trend that was also seen in the grasping task.

---

> ### Author Response · Authors · 2025-11-22
> **Response to Reviewer CYqi, Part 2**
>
> **3. Discussion of Related Work:** In addition, we outline below how our method differs from GROOT, DexGraspVLA, and other object-centric approaches. We will add these comparisons in the final version of the paper as well.
>
> - **GROOT (Learning Generalizable Manipulation Policies with Object-Centric 3D Representations):** GROOT is an object-centric approach that predicts robot actions from object point clouds. Like LEGO, it tracks task-relevant objects using masks; however, while GROOT relies on backprojected 3D point clouds, LEGO uses masks to constrain attention in the vision encoder. Although point clouds enable generalization to structurally similar objects, they often struggle with completely new shapes, as shown by the results in the table below. In contrast, LEGO demonstrates stronger generalization—transferring directly from randomized toys to real-world objects—driven by the object-centric representation from detection pooling. We include GROOT as a baseline in our ManiSkill YCB benchmark as detailed above.
>
> - **Transferring Foundation Models for Generalizable Robotic Manipulation:** This approach leverages VLMs and SAM to generate segmentation masks, which are fed as additional inputs to the policy. While sharing our goal of focusing on the target, a key distinction lies in the architecture: this method treats the mask as an extra input channel, whereas LEGO uses it to explicitly set the attention mask in the vision encoder. Furthermore, while this prior work demonstrates generalization within similar real-world object sets, LEGO achieves a more extreme form of zero-shot generalization. By creating a stronger object-centric representation, our approach transfers directly from randomized toys to real-world objects, bridging a significantly larger domain gap.
>
> - **DexGraspVLA** is a VLA model employing a planner-controller framework. It utilizes a large pre-trained VLM (Qwen2.5-VL-72B-Instruct) to decompose prompts into bounding-box instructions, which guide a diffusion controller. While sharing our object-centric reliance on target masks, key differences distinguish our approaches. DexGraspVLA depends on a massive VLM with internet-scale knowledge for high-level planning. In contrast, our model is orders of magnitude smaller, focusing specifically on grasping to demonstrate that the visual representation induced by detection pooling drives object-level generalization. We demonstrate that our approach offers a strong inductive bias that promotes learning efficiency, and view integrating detection pooling with pre-trained VLMs as a promising direction to extend this capability to broader manipulation tasks.
>
> - **Object-Centric Representations Improve Policy Generalization in Robot Manipulation:** This work evaluates a variety of visual encoders, demonstrating that object-centric representations significantly enhance generalization, particularly under visual distribution shifts. These findings further validate the premise of our detection pooling mechanism: by focusing the model on grasping-relevant features, we enable effective zero-shot transfer to novel objects
>
> - **DOCIR (Disentangled Object-Centric Image Representation for Robotic Manipulation):** This work demonstrates that decomposing the scene into separate visual representations improves generalization. Specifically, it uses segmentation masks to generate and individually encode distinct images for the target object, robot, and obstacles. While LEGO shares the use of masks to focus on the target, it differs architecturally: rather than processing separate image inputs, LEGO uses the mask to constrain attention within the vision encoder. Experimentally, our policy demonstrates substantially stronger generalization; while DOCIR evaluates on just two object categories, LEGO successfully transfers from randomized toys to a wide variety of diverse real-world objects.

---

### Author Response · Authors · 2025-11-22
**High-Level Summary of the Rebuttal**

We thank the reviewers for their thoughtful and constructive feedback. In this work, we introduce LEGO, an object-centric manipulation framework that achieves zero-shot generalization from a small set of randomized toys to real-world objects through the simple yet powerful detection pooling mechanism.

We are encouraged that reviewers highlighted LEGO’s **“strong empirical effect”** (`oAEF`) with **“minimal data”** (`V8CY`), described the use of randomized toys as **“insightful”** (`CYqi`), and found the overall motivation **“clear and compelling”** (`oAEF`). Reviewers also emphasized our **“extensive ablations”** (`oJDK`) and the contribution of **“concrete evidence for object-centric representations in embodied manipulation”** (`7VRd`). We sincerely appreciate these positive assessments and have strengthened the paper further in response to reviewer suggestions.
To incorporate reviewers’ feedback, we made the following key revisions:

1. **Comparison with object-centric approaches.** We added both experimental and qualitative comparisons to recent object-centric baselines (reviewers `CYqi`, `V8CY`). The new results show that our model consistently outperforms these methods, and the qualitative analysis highlights conceptual differences that explain this gap.

2. **New Manipulation Task.** To address Reviewer `CYqi`'s suggestion and demonstrate the generality of detection pooling, we introduced a new 'slide object to target' task. This task demands precise contact and controlled sliding, distinct from grasping. Our results show that LEGO significantly outperforms the OpenVLA-OFT baseline in this setting, reinforcing the robustness of our approach.

3. **Effect of mask noise.** As suggested by reviewers `oAEF`, `oJDK`, `7VRd`, and `V8CY`, we added an experiment evaluating robustness to noisy/offset masks. Even with substantial noise, LEGO exhibits minimal performance degradation, underscoring its tolerance to imperfect segmentation.

4. **Robustness analysis.** As suggested by reviewers `oAEF` and `7VRd`, we added extensive experiments studying the effects of lighting changes, clutter, and lookalike distractors. Across all cases, LEGO demonstrates strong resilience and generalization due to detection pooling.

5. **Scaling curve and control-mode ablations.** We have added two more data points between 1 and 25 toys to Figure 4 of the original paper, for 5 and 15 unique toy sets, as suggested by reviewer `oJDK`. In addition, as reviewer `oJDK` suggested, we also compare absolute and delta joint control and empirically show that absolute control leads to better performance for our model.

6. **Crop-box baseline.** Following reviewer `oAEF`, we added a crop-box baseline, which confirms that naively cropping to the object region is insufficient as visual input and verifies the importance of detection pooling.

7. **Context-length study.** As requested by reviewer `7VRd`, we conducted a detailed ablation on condition length (C) and prediction length (K), revealing how temporal context affects performance and offering guidelines for selecting these hyperparameters.

We next address all reviewers’ concerns separately and look forward to an open and constructive discussion. We will also add these changes to the final version of the paper.

---

### Author Response · Authors · 2025-12-02
**Summary of Rebuttal after Area Chair Reassignment**

We would like to thank all reviewers for their feedback thus far. In light of the area chair reassignment, we would like to take this opportunity to briefly summarize the status and updates of our paper.

In this work, we introduce **LEGO**, an object-centric manipulation framework that achieves **zero-shot generalization from a small set of randomized toy objects to real-world settings** through the simple yet powerful DetPool mechanism. Below, we provide a high-level summary of each review and our rebuttal response.

`Reviewer CYqi` found the idea of using randomized toys “insightful”, and the significant number of real-world robot experiments as “excellent”. In our rebuttal, we addressed this reviewer’s suggestions in the following ways:

1. **Related work comparison.** We added both experimental and qualitative comparisons to suggested object-centric baselines. The new results show that our model consistently outperforms these methods, and the qualitative analysis highlights conceptual differences that explain this gap.

2. **New manipulation task.** To demonstrate that DetPool is a general approach that can work across diverse manipulation tasks, we introduced a new “slide object to target” task, which requires precise contact and controlled sliding. LEGO significantly outperforms the OpenVLA-OFT baseline, reinforcing the robustness of our approach.

`Reviewer oAEF` found our motivation “clear and compelling”, with our results showing “strong empirical effect” and “potential for a strong contribution”. This reviewer also highlighted that they would *"be glad to reconsider my evaluation if you can effectively address the issues highlighted in the review and outline a concrete plan to strengthen the experimental validation."* In our rebuttal, we addressed this reviewer’s suggestions in the following ways:

1. **Object-centric VLA baselines:** We added object-centric versions of our $\pi_0$-FAST and OpenVLA-OFT baselines.

2. **Mask noise and robustness experiments:** We added comprehensive experiments studying the effect of mask noise, lighting and distractors.

`Reviewer oJDK` found our work to have “positive results” and “extensive ablations”. In our rebuttal, we addressed this reviewer’s suggestions in the following ways:

1. **5, 15 toy experiments:** We added new data points to study the performance between 1 and 25 unique toys, as recommended.

2. **Delta vs absolute control:** We added a detailed experiment studying the effect of using delta vs absolute control.

3. **Masking constraints:** We added experiments involving clutter and mask noise to analyze the robustness of our model.

4. **Small clarifications:** This reviewer also asked some small clarification questions like physical properties of objects and analysis of failure modes, each of which we have answered in detail.

`Reviewer 7VRd` emphasized the “concrete evidence for object-centric representations in embodied manipulation” and our work’s “broad impact”. This reviewer also highlighted that *"if the authors can address these questions convincingly during rebuttal, I am inclined to raise my score."* In our rebuttal, we addressed this reviewer’s suggestions in the following ways:

1. **Mask noise and robustness of the method:** We have added comprehensive experiments studying the effect of mask noise, which demonstrate the robustness of our model.

2. **Significant difference of the proposed DetPool with RoI-guided pooling and mask-guided token selection:** We add further clarifications and also ablate the mask-guided token selection to verify the rationale and effectiveness of the proposed DetPooling.

3. **Condition and prediction length:** As suggested, we conducted an extensive ablation study to analyze the effect of changing condition and prediction lengths, validating our choices of C=16, K=16.

4. **Geometric similarity:** As suggested by the reviewer, we have added a quantitative comparison between the training and test set objects using the chamfer distance metric, and also found the success rate by using similarity buckets. The results show that our model is robust to more complex and unfamiliar object shapes as well.

`Reviewer V8CY` found our work to have “strong empirical performance with minimal data”, with the data efficiency claim being “well demonstrated empirically”. In our rebuttal, we addressed this reviewer’s suggestions in the following ways:

1. **Robustness experiment:** We have added detailed experiments studying the effect of lighting, clutter and distractors on our model, which remains robust under these conditions.

2. **Comparison with other object-centric approaches:** We add comparisons to the suggested object-centric works.

3. **Response to approach being heuristic and not supporting toy-based generalization:** Our response details how our approach clearly supports toy-based generalization, and also addresses the question of considering physical properties of the object such as mass.

Best,

Authors

---

### Meta-Review · Area_Chair_suah · 2026-01-04

**Summary:**

I recommend accepting the paper. Reviewers generally appreciated the idea presented and careful experiments confirming the claims. Manipulation is a difficult problem to crack. The paper brings a novel and simple perspective, and the authors responded to reviewers with focused effective rebuttals.

Note to authors: The paper does not state the real-time duration of one timestep, so seconds-per-step and steps-per-grasp are not precisely specified.

**Reviewer Concerns:**

Main concerns:

Reviewer CYqi:
- Detection pooling approach is taken in other works and should be discussed.
- - Additional comparison added.
- Evaluation is limited to grasp tasks, not other contact-rich manipulation tasks.
- - More tasks added

Reviewer oAEF:
- Inaccurate/inconsistent implementation detail
- - Author acknowledged and seems not damaging
- Using object mask only for new approach makes comparison unfair
- - Authors provided proper experiments as a response

Reviewer oJDK’s moderate concerns and already has positive impression:
- Does not capture desired grasp location
- - Authors acknowledges that it is not the focus, which is reasonable
- Does not capture special physical properties such as delicate parts
- - Authors acknowledges that it is not the focus, which is reasonable
- Masking step may not extend to complex scenarios with multiple objects or occlusion
- - Additional experiments with clutter added to show robustness

Reviewer 7VRd (positively inclined):
- Mask dependence and robustness questioned
- - Additional experiments on robustness provided
- Evaluation too idealized, focusing on single objects
- - Additional experiments with clutter provided
- Insufficient ablation
- - Additional ablation provided

Reviewer V8CY:
- Questions generalization when geometry is not enough
- - Authors agree and leave other forms of generalization for the future work while sticking with the claim that visual generalization does a lot.

**Reviewer Scores:**

Given the extensive responses and additional experiments during rebuttal, I would have moved the slightly negative scores to positive ones.

---

### Decision · Program_Chairs · 2026-01-26

Accept (Poster)